# Development of antibacterial compounds that constrain evolutionary pathways to resistance

**Yanmin Zhang[1,2†], Sourav Chowdhury[2†], João V Rodrigues[2], Eugene Shakhnovich[2*]**

[1]School of Science, China Pharmaceutical University, Nanjing, China; [2]Department of Chemistry and Chemical Biology, Harvard University, Cambridge, United States

**Abstract** Antibiotic resistance is a worldwide challenge. A potential approach to block resistance is to simultaneously inhibit WT and known escape variants of the target bacterial protein. Here, we applied an integrated computational and experimental approach to discover compounds that inhibit both WT and trimethoprim (TMP) resistant mutants of *E. coli* dihydrofolate reductase (DHFR). We identified a novel compound (CD15-3) that inhibits WT DHFR and its TMP resistant variants L28R, P21L and A26T with $IC_{50}$ 50–75 μM against WT and TMP-resistant strains. Resistance to CD15-3 was dramatically delayed compared to TMP in in vitro evolution. Whole genome sequencing of CD15-3-resistant strains showed no mutations in the target folA locus. Rather, gene duplication of several efflux pumps gave rise to weak (about twofold increase in $IC_{50}$) resistance against CD15-3. Altogether, our results demonstrate the promise of strategy to develop evolution drugs - compounds which constrain evolutionary escape routes in pathogens.

## Introduction

Fast-paced artificial selection in bacteria against common antibiotics has led to the emergence of highly resistant bacterial strains which potentially render a wide variety of antibiotics clinically ineffective. Emergence of these 'superbugs' including ESKAPE (*Enterococcus faecium*, *Staphylococcus aureus*, *Klebsiella pneumoniae*, *Acinetobacter baumannii*, *Pseudomonas aeruginosa*, and *Enterobacter spp.*) (*Peneş et al., 2017*) call for novel approaches to design antibiotic compounds that act as 'evolution drugs' by blocking evolutionary escape from antibiotic stressor. Currently, drug development protocols and treatment strategies have limited success in addressing the issue of drug resistance as it overlooks the driving complex underlying evolutionary selection processes. The problem of drug resistance and its potential solutions constitutes important aspects of 'Evolutionary medicine' (*Stearns, 2012*; *Williams and Nesse, 1991*). Apart from addressing specific questions involving disease mechanism and their ontogeny, evolutionary medicine takes a more comprehensive approach and aims to address evolutionary questions such as selective advantage of a phenotype and/or the associated phylogeny (*Nesse and Stearns, 2008*). This makes it immensely pertinent in the context of the problem of drug resistance which is primarily an evolutionary escape under conditions of selection pressure exerted by chemical stressors viz. antibiotics. Application of evolution principles has been promising in the studies of infectious diseases, aging, and cancer therapy (*Nesse and Stearns, 2008*). Evolution of cancer cells and their complex ways to evade the anti-cancer therapeutics in many ways resembles the evolutionary selection of resistant bacterial cells evading the antibiotic action (*Stearns, 2012*). Development of 'evolution drugs' – *compounds that block or slow down escape routes to resistance* – is inspired by recent progress in understanding of evolutionary dynamics of pathogen escape from stressors (*Chakraborty and Barton, 2017*; *Gong et al., 2013*; *Klein et al., 2018*; *Marchi et al., 2019*; *Rotem et al., 2018*). Recent report of compounds modulating proteins controlling mutation rates is an interesting lead in that direction (*Ragheb et al., 2019*). Using evolution drugs to slow

**\*For correspondence:**
shakhnovich@chemistry.harvard.edu

[†]These authors contributed equally to this work

**Competing interest:** The authors declare that no competing interests exist.

down evolutionary dynamics of pathogen or cancer escape can help tilt the balance of evolutionary arms-race in favour of the host immune system.

Selectively targeting bacterial proteins which are critical to essential bacterial life processes like cell wall biosynthesis, translation, DNA replication etc. with novel compounds forms the basis of antibiotics development programs. Dihydrofolate reductase (DHFR) is one such protein, which, due to its critical role in nucleotide biosynthesis, has been a central antibacterial and anticancer drug target (*Lin and Bertino, 1991*; *Schweitzer et al., 1990*). Based on the chemical scaffold, DHFR inhibitors can be divided into classical and non-classical ones. The classical DHFR inhibitors generally contain a 2,4-diamino-1,3-diaza pharmacophore group (*Srinivasan et al., 2017*) which constitute structural analogues of its substrate dihydrofolate and competitively bind the receptor-DHFR active site. Inhibitors of this type such as methotrexate (MTX) (*Bleyer, 2015*) and pralatrexate (PDX) (*Izbicka et al., 2009*) are approved as anticancer drugs. In addition, predominant classes of inhibitors derived from dihydrofolate analogues also include diaminoquinazoline, diaminopyrimidine, diaminopteridine, and diaminotriazines (*Srinivasan et al., 2017*). Non-classical antifolate drugs like trimethoprim (TMP) (*Finland and Kass, 1973*) trimetotrexate (TMQ) (*Lin and Bertino, 1991*), that interact selectively with bacterial but not human DHFR are approved as antibacterial drugs. Without the solvent accessible group of glutamic acid, they are more fat-soluble, passively diffuse into cells, and are also not substrates for folylpolyglutamate synthetase enzymes. However, due to rapid emergence of resistant mutations in DHFR, the development of drug resistance to antifolate antibiotics belonging to any of the above-mentioned classes presents a significant challenge (*Huovinen et al., 1995*). Nevertheless, TMP and other non-classical anti-folates can be attractive templates for bringing in modifications in the structure of DHFR inhibitors and derive novel DHFR inhibitors (*Wróbel et al., 2020*).

In general, both clinical and in vitro studies have shown that accumulation of point mutations in critical amino acids residues of the binding cavity represent an important mode of trimethoprim resistance. Factors associated with TMP resistance are far more complex in the clinical isolates. A recent study has shown the contribution of mobile dfrA genes to TMP resistance in the emerging pathogen Acinetobacter baumannii and its association to chromosomal folA genes in rapidly mobilizing novel mutations (*Sánchez-Osuna et al., 2020*). The study shows how sulfonamide resistance in general extends to TMP with generalized chromosomal resistance determinants predating the origin of several genera and several clusters of resistance genes disseminated broadly among clinical isolates. Further, whole-genome sequencing of TMP-resistant *E. coli* clinical isolates showed the contribution of dfrB genes toward clinical TMP resistance (*Toulouse et al., 2017*). Thus, mechanisms leading to TMP resistance could be extremely diverse and complex. These complications notwithstanding, in our current study we focus on a simplified experimental model system of in vitro evolution whereby resistant phenotypes emerge from point mutations in the folA locus as outlined in detail below.

Point mutations conferring resistance in bacteria to anti-DHFR compounds are primarily located in the folA locus that encodes DHFR in *E. coli* (*Oz et al., 2014*; *Tamer et al., 2019*; *Toprak et al., 2012*) making DHFR an appealing target to develop evolution antibiotic drugs. A possible approach is to design compounds that can inhibit the wild type (WT) DHFR along with its resistant variants thus making multiple evolutionary pathways toward drug-resistance less accessible. In this work, we developed an integrative computational modeling and biological evaluation workflow to discover novel DHFR inhibitors that are active against WT and resistant variants. Structure-based virtual screening (SBVS) including molecular docking with subsequent validation by molecular dynamics (MD) (*Cheron and Shakhnovich, 2017*; *Leonardo et al., 2015*; *Liu et al., 2020*; *Zhang et al., 2017*) were used to screen a large compound database. We found a series of DHFR inhibitors with novel scaffolds that are active against both the WT and several mutant DHFR proteins and are cytotoxic against WT *E. coli* along with *E. coli* strains with chromosomally incorporated TMP-resistant DHFR variants (*Palmer et al., 2015*). Those inhibitors are more potent against the escape variants than the WT DHFR. This makes them promising candidates for further development of next generation of antibiotics that prevent fast emergence of resistance.

## Results

### In silico search for potent broad DHFR inhibitors

The key objective of our approach is to find compounds that *simultaneously* inhibit WT and drug resistant variants of a target protein. Firstly, we developed an integrative computational workflow including molecular docking, molecular dynamics and evaluation of protein-ligand interaction along with Lipinski's rule of five (*Manto et al., 2018*) filter to screen two commercial databases that include about 1.8 million compounds (*Figure 1A*). First, we assessed which conformation of M20 loop of DHFR (closed, open or occluded) should be used for molecular docking. To that end, we evaluated which conformation of the M20 loop in the target structure gives rise to best agreement between docking score and experimental binding affinity for known DHFR inhibitors. By classification (see Materials and methods: *Crystal structure selection*), we selected four representative crystal structures (closed: PDB 1*R*X3, open: PDB 1RA3, occluded: PDB 1RC4 and PDB 5CCC, *Figure 1—figure supplement 1*) as putative target structures for docking. Using the closed conformation of M20 loop (PDB 1*R*X3), we were able to recover the largest proportion of known inhibitors (Table 2 and *Figure 1—figure supplements 2 and 3*). Therefore, the closed conformation of M20 loop (PDB 1*R*X3) was adopted as the most representative crystal structure for the initial SBVS of compound databases for novel broad range inhibitors. It was also used for subsequent in-depth evaluation of most promising candidates using molecular dynamics simulation. The detailed discussion of the rationale behind selection of closed conformation is provided in the Materials and methods: *Crystal structure selection*.

A total of 307 candidate compounds with strongest docking score that form hydrogen bond with the critical residue Asp27 in the DHFR-binding pocket, were submitted for more accurate prediction of binding free energy (*Cheron and Shakhnovich, 2017*; *Figure 1A*). Our approach to predict binding free energy is based on a series of relatively short MD simulations of binding conformations with subsequent MMPBSA scoring as presented in *Cheron and Shakhnovich, 2017*. Next, we assessed the accuracy of this approach for WT and mutant DHFR in reproducing binding affinities of known ligands. To that end, we built linear regression equation models (see Figure 1B and Figure 1C and Materials and methods: *Construction of binding affinity prediction model*) to predict binding free energies calculated by MMPBSA.

The models reproduced known binding affinities with high accuracy (see *Figure 1B and C* and *Figure 1—figure supplement 4* and *Figure 1—figure supplement 6*). Additionally, we constructed linear regression equation models to predict binding free energy of TMP against WT DHFR from Listeria grayi (*L. grayi*) and Chlamydia muridarum (*C. muridarum*) again showing highly significant correlation between predicted and experimental values (*Figure 1—figure supplement 7*), demonstrating broad predictive power of the method. More detail on the construction of binding affinity prediction models can be found in the Materials and methods: *Construction of binding affinity prediction model*.

Further, the analysis of the DHFR crystal structures showed that Asp27 forms hydrogen bond with almost all DHFR inhibitors in the ligand-binding cavity. Thus, compounds predicted by docking that make hydrogen bond with Asp27 (*Figure 1A*) and having MMPBSA predicted binding free energies less than –20 kcal/mol (*Supplementary file 1*. *Compound Information*) against both the WT and all TMP-resistant variants of DHFR were selected for further evaluation. Out of this set, we selected the novel compounds that differ substantially from 183 known DHFR inhibitors (see Materials and methods: *Selection of virtual screening hits*). Generally, compounds with similar properties tend to have similar activity (*Kumar, 2011*). Based on that principle, we compared two-dimensional (2D) physicochemical properties (*Zhang et al., 2013*) and protein-ligand interaction fingerprint (PLIF) features (*Marcou and Rognan, 2007*) of the prospective set with those of the known inhibitors. Results showed that the selected hits have high similarity in both 2D-physicochemical properties and PLIF features when compared with known DHFR inhibitors, which suggest their potential inhibitory activity against DHFR (*Figure 1—figure supplement 8* and *Figure 1—figure supplement 9*). On the other hand, they show relatively low chemical similarity (*Figure 1—figure supplement 8*) with the known DHFR inhibitors, suggesting that the selected hits are chemically novel. Altogether, a total of 40 prospective active compounds were purchased for evaluation. The detailed information on all compounds can be found in *Supplementary file 1*. (*Compound Information*). Further details on SBVS can be found in the Method and Materials: *Selection of virtual screening hits*.

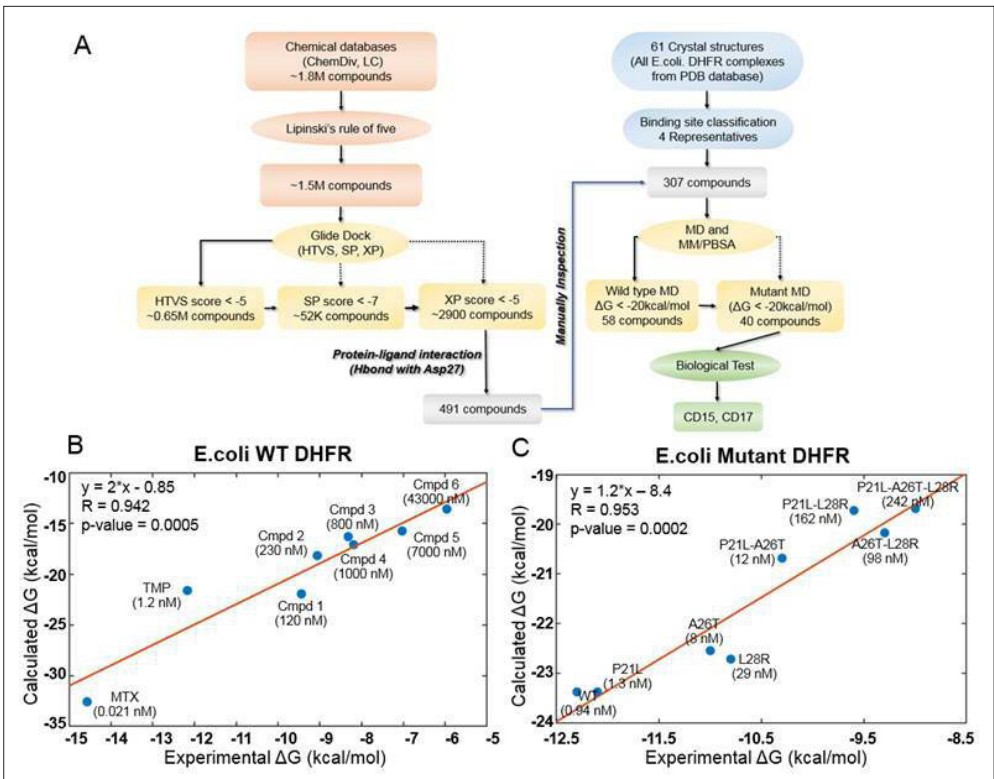

**Figure 1.** Computational design of broadly neutralizing DHFR inhibitors effective against WT and resistant DHFR mutant strains. (**A**) Integrative virtual screening workflow. Detailed description of the virtual screening workflow can be found in Method and Materials: Selection of virtual screening hits. (**B**) Linear model for binding affinity prediction constructed using known binding affinities of eight known inhibitors of WT *E. coli* DHFR (***Figure 1—figure supplement 4***) obtained from ***Carroll et al., 2012***. (**C**) Linear model for binding affinity prediction constructed using experimental inhibitory activity for TMP against WT DHFR and seven resistant DHFR mutants (***Rodrigues et al., 2016***). MD simulation and MM/PBSA affinity evaluation protocol (***Cheron and Shakhnovich, 2017***) was applied to calculated binding free energy of complexes of *E. coli* DHFR with eight known inhibitors and the calculated values were compared with the reported experimental binding affinities (Kd or Ki values).

The online version of this article includes the following figure supplement(s) for figure 1:

**Figure supplement 1.** Three different types of M20 loops in *E. coli* DHFR.

**Figure supplement 2.** The number distribution of inhibitors with a given XP docking score.

**Figure supplement 3.** The scatter plot experimentally measured activities of DHFR inhibitors vs their XP docking score with various DHFR conformations as targets.

**Figure supplement 4.** Compounds used for building the binding affinity prediction model.

**Figure supplement 5.** Comparison of docked TMP in *E. coli* DHFR (cyan, PDB 1RX3) with crystalized TMP with *Staphylococcus aureus* DHFR (salmon, PDB: 2W9G).

**Figure supplement 6.** Linear correlation between the computational and experimental binding Gibbs-free energies for eight compounds against WT *E. coli* DHFR (upper panel).

**Figure supplement 7.** Correlation and linear models for the calculated and experimental binding Gibbs-free energies for TMP against WT and mutant Listeria grayi (upper panel) and *Chlamydia muridarum* DHFR (lower panel).

**Figure supplement 8.** Similarity comparison between the selected hits with the known inhibitors using physicochemical properties, structure (represented by ECFP4) and PLIF. ChemDiv hits and LC hits represent hit compounds screened from ChemDiv and Life Chemicals database, respectively.

**Figure supplement 9.** Two-dimensional chemical space of physiochemical properties for the selected hits with the known DHFR inhibitors.

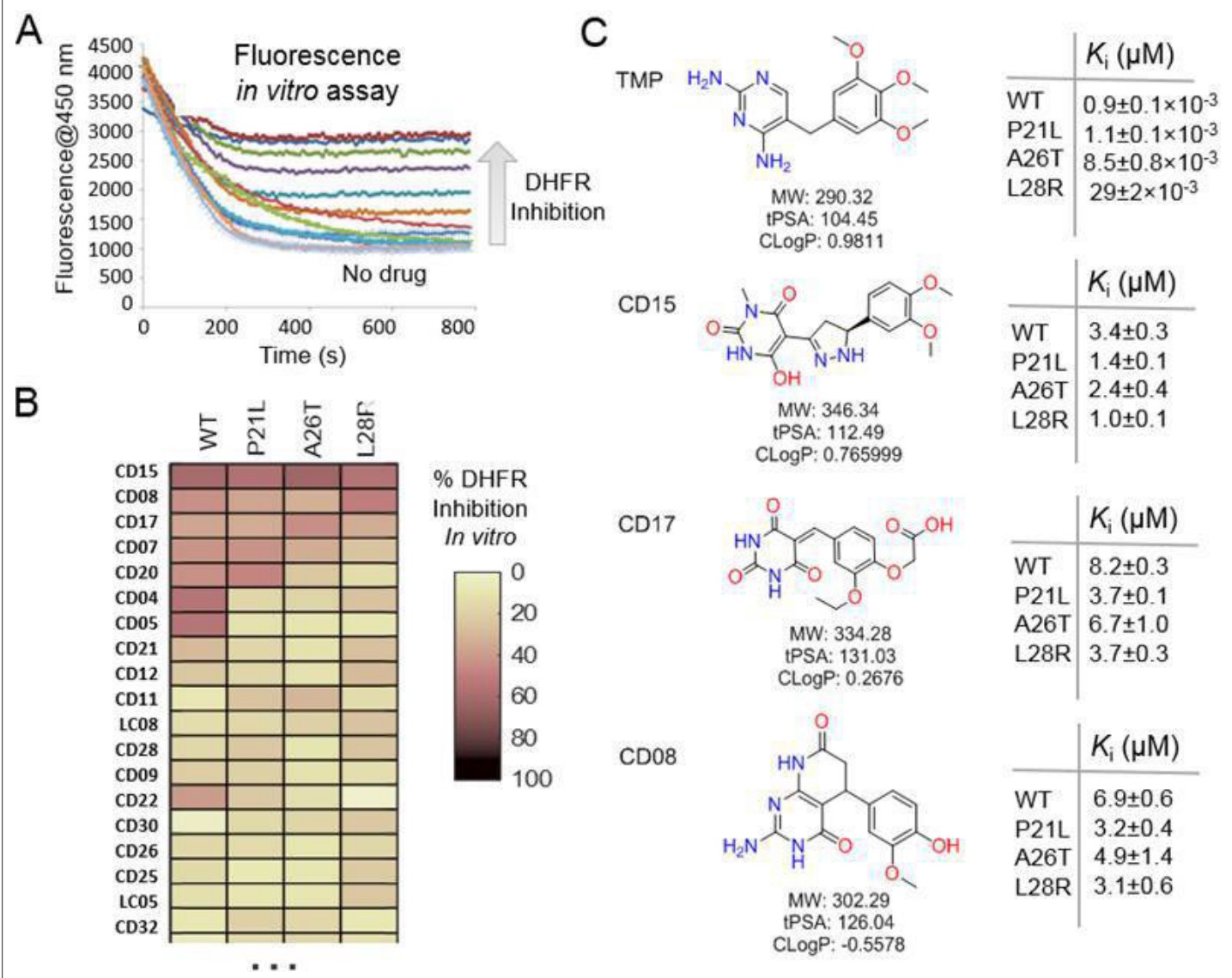

**Figure 2.** Evaluation of the potential hits in vitro and their optimization. (**A–B**) An in vitro kinetic assay of DHFR catalytic activity was used to screen inhibitors against WT DHFR and three single mutants resistant to TMP (P21L, A26T and L28R). (**C**) Chemical structures of the top three compounds showing simultaneously the highest potency against WT and mutant DHFR variants. The structure of trimethoprim is shown for comparison.

The online version of this article includes the following figure supplement(s) for figure 2:

**Figure supplement 1.** The initial inhibition rate of catalytic activity of the selected 40 hits against WT and three DHFR mutants at a single concentration of 200 µM.

## Assaying prospective compounds in vitro

A spectrophotometric assay (see Materials and methods) was employed to evaluate possible inhibition of catalytic activity of the 40 selected compounds against WT DHFR and its TMP resistant mutants including P21L, A26T, and L28R (*Figure 2*). All 40 prospective compounds were initially assayed for inhibition of DHFR at a single fixed concentration of 200 µM. As shown in *Figure 2A-B* and *Figure 2—figure supplement 1*, a total of 13, 8, 6, and 14 compounds resulted in more than 20% loss of the DHFR catalytic activity at that concentration for WT, P21L, A26T, and L28R DHFR, respectively. Among them, compounds CD15, CD17, and CD08 showed more than 30% inhibition against both WT and all three DHFR mutants (*Figure 2C* and *Figure 2—figure supplement 1*). Compound CD20, with similar scaffold to that of TMP, showed inhibition against WT, P21L, and A26T, but not L28R DHFR and

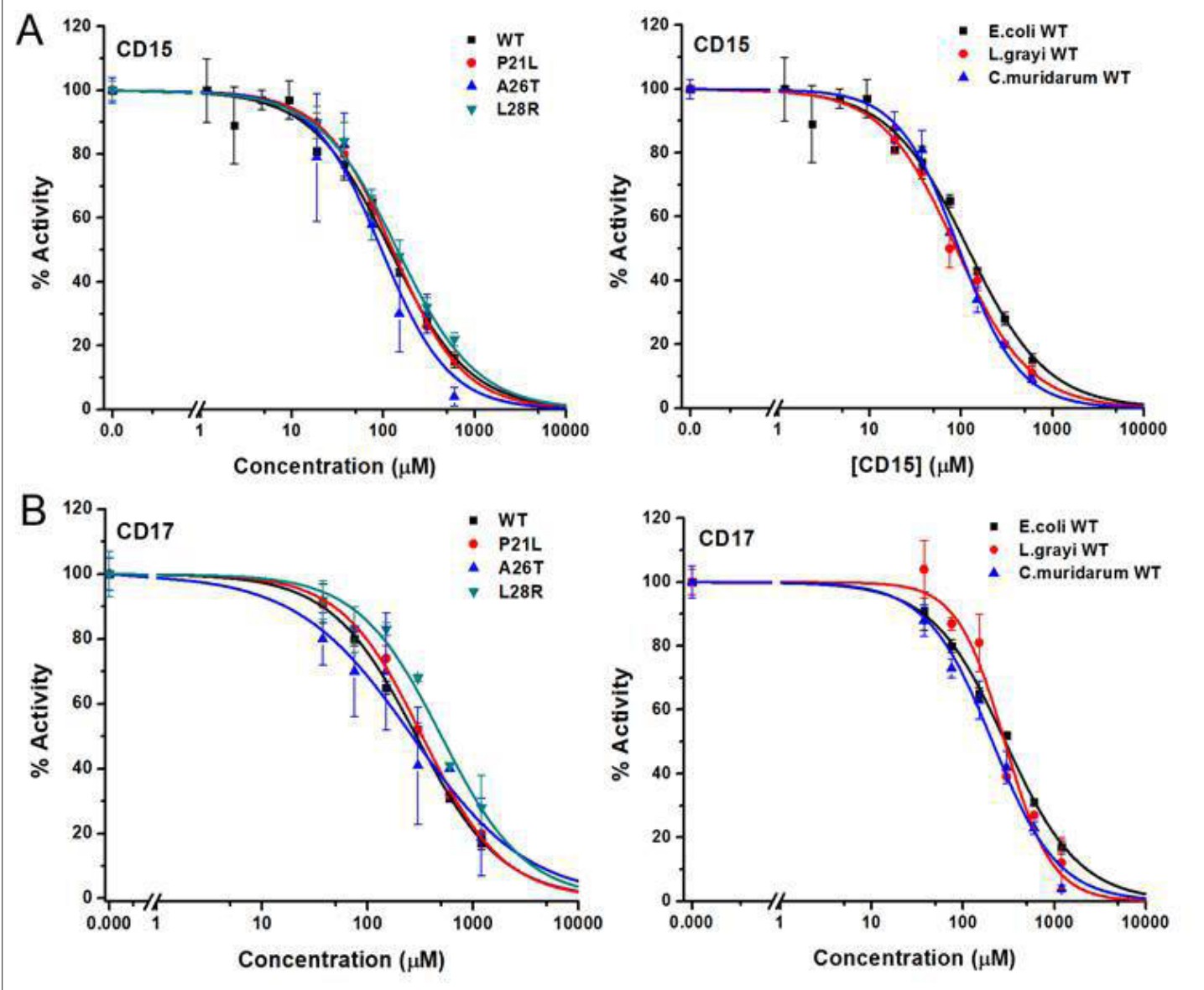

**Figure 3.** Inhibition of WT and mutant DHFR from different species by CD15 and CD17. (**A**) Concentration-dependent inhibition curves for compound CD15 for WT and mutant DHFR of *E. coli* (left panel) and for WT DHFR from *E. coli*, *L.grayi*, and *C.muridarum*, respectively (right panel). (**B**) Concentration-dependent inhibition curves for compound CD17 on WT and mutant DHFR of *E. coli* (left panel) and of WT DHFR from *E. coli*, *L. grayi*, and *C. muridarum*, respectively (right panel). The %Activity of the y-axis is represented by the decrease of fluorescence at 450 nm for the reaction system (see Materials and methods for more detail).

was no longer considered. Thus, a total of three hits including compounds CD15, CD17 and CD08 (*Figure 2C*) were further evaluated for concentration-dependent inhibition of all DHFR variants.

Two of the three compounds, CD15 and CD17 with two novel scaffolds, inhibited WT and mutant DHFRs in concentration-dependent manner (*Figure 3*). We used inhibition curves (*Figure 3*), to obtain $IC_{50}$ values and converted them into corresponding Ki values (*Table 1*). For compound CD15, the Ki values were all less than 5 µM against WT and three single-point DHFR mutants. For the L28R mutant, the Ki values were 1.0 µM, outperforming the WT, P21L, or A26T mutant DHFR. It is worth noting that L28R is a strong TMP escape variant (*Rodrigues et al., 2016*; *Toprak et al., 2012*), thus the discovered CD15 series appeared a promising candidate for an 'evolution drug' that has a potential to suppress even most intractable escape variants. Based on these data we decided to proceed with compounds CD15 and CD17 for in depth evaluation. In addition, we evaluated activity of the two

**Table 1.** The Ki values (in µM) for compounds CD15 and CD17.

| Species | DHFR type | CD15 Ki value | STD* | CD17 Ki value | STD | CD15-3 Ki value | STD | TMP Ki value | STD |
|---|---|---|---|---|---|---|---|---|---|
| *E. coli* | WT | 3.35 | 0.28 | 8.18 | 0.29 | 5.52 | 0.98 | $0.90 \times 10^{-3}$ | $0.10 \times 10^{-3}$ |
| | P21L | 1.42 | 0.04 | 3.70 | 0.13 | 2.51 | 0.31 | $1.10 \times 10^{-3}$ | $0.10 \times 10^{-3}$ |
| | A26T | 2.43 | 0.37 | 6.73 | 1.02 | 3.48 | 0.18 | $8.50 \times 10^{-3}$ | $0.80 \times 10^{-3}$ |
| | L28R | 1.04 | 0.07 | 3.65 | 0.33 | 0.98 | 0.11 | $29.0 \times 10^{-3}$ | $2.00 \times 10^{-3}$ |
| | P21L-A26T | 3.26 | 0.21 | 9.20 | 0.71 | | | | |
| | P21L-L28R | 0.56 | 0.05 | 1.37 | 0.06 | | | | |
| | A26T-L28R | 0.61 | 0.04 | 2.04 | 0.11 | | | | |
| *L. grayi* | WT | 5.01 | 0.29 | 16.17 | 1.70 | | | | |
| *C. muridarum* | WT | 14.60 | 0.64 | 32.38 | 2.53 | | | | |
| *Human* | WT | 0.38 | 0.05 | 0.74 | 0.10 | NT† | | | |

*STD means the standard error from three duplicate experiments.
†NT means no inhibition was detected at the maximal compound concentration tested.

most promising inhibitors, CD15 and CD17 against WT DHFR from two more species: *L.grayi* and *C.muridarum*. We found that both compounds showed similar inhibition against the two species (see *Figure 3* and *Table 1*). These results suggest the potential of the two compounds as broadly efficient potential antibacterial leads.

We also evaluated inhibitory activity of CD15 against double resistant mutants and found that they are approximately as active or better than against single mutants (see *Table 1*).

## Broad antimicrobial activity of new compounds

Since two of the 40 compounds inhibit both WT and mutant proteins in vitro, we proceeded to test their efficacy to inhibit growth of *E. coli*. Previously (*Rodrigues et al., 2016*), we used strains with chromosomal replacement of WT folA with folA gene encoding three single mutants including P21L,

**Table 2.** The IC$_{50}$ values for the in vivo inhibition of several CD15 and CD17 series compounds.

| DHFR | Bacterial growth inhibition IC$_{50}$ (µM) | | | | | | | | | | | |
|---|---|---|---|---|---|---|---|---|---|---|---|---|
| | CD15 | STD* | CD15-2 | STD | CD15-3 | STD | CD15-4 | STD | CD15-6 | STD | TMP | STD |
| WT | 170 | 12 | 129 | 22 | **72** | 6 | 302 | 74 | 175 | 4 | 1.3 | 0.4 |
| P21L | 176 | 10 | 125 | 24 | **57** | 7 | 92 | 15 | 159 | 6 | 12 | 2 |
| A26T | 197 | 18 | 119 | 11 | **73** | 6 | 123 | 25 | 172 | 4 | 3.9 | 0.7 |
| L28R | 159 | 14 | 88 | 6 | **48** | 5 | 67 | 4 | 167 | 10 | 114 | 7 |

| DHFR | Bacterial growth inhibition IC$_{50}$ (µM) | | | | | |
|---|---|---|---|---|---|---|
| | CD17 | STD | CD17-3 | STD | CD17-4 | STD |
| WT | 1,774 | 22 | 2,946 | 967 | ND† | ND |
| P21L | 5,048 | 809 | 2,845 | 437 | ND | ND |
| A26T | 1920 | 135 | 2,152 | 265 | ND | ND |
| L28R | 932 | 44 | 907 | 57 | 1,857 | 272 |

*STD means the standard error from three duplicate experiments.
†ND (not determined) indicates no result was obtained for the compound against WT and three mutant DHFR.

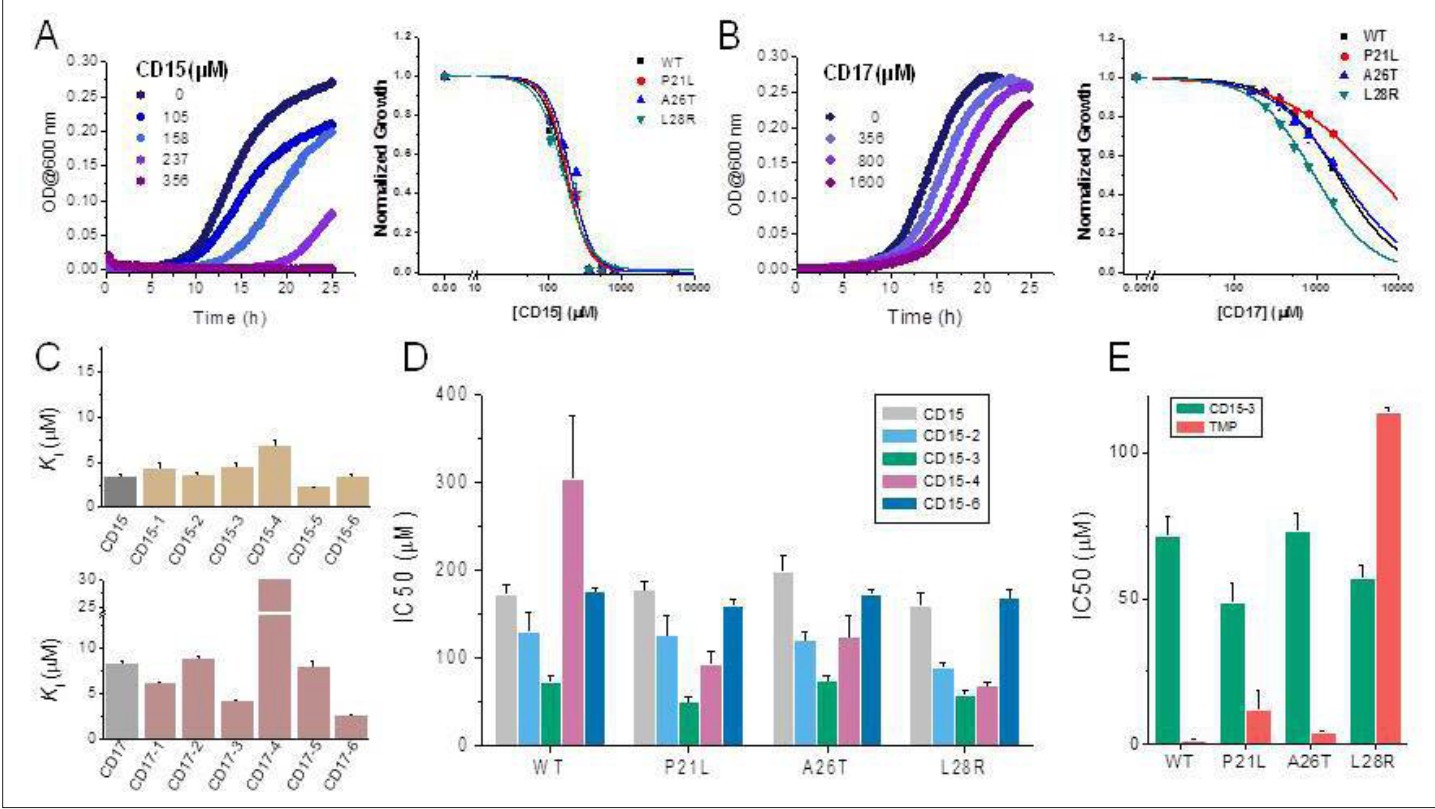

**Figure 4.** Compounds CD15 and CD17 inhibit growth of WT and TMP-resistant mutant *E. coli* strains. (**A**) Growth curves for WT strain (left panel) at different concentrations of CD15 and normalized (by maximal growth in the absence of stressor) inhibition by CD15 curves for WT and various TMP-resistant DHFR mutants. (**B**) same as A for CD17. Measurements in the presence of different drug concentrations were performed in a 96-well microplate at 37°C. (**C**) Optimization of compounds CD15 (upper panel) and CD17 (lower panel) lead to hits with increased in vitro inhibitory potency toward WT *E. coli* DHFR. (**D**) The CD15 series compounds inhibit growth of WT and resistant mutant *E. coli* strains. For each strain, growth measurements were performed in the presence of varying concentrations of compounds. (**E**) Comparison of $IC_{50}$ of inhibition of growth of WT and TMP-resistant mutant *E. coli* strains for CD15-3 and TMP.

The online version of this article includes the following figure supplement(s) for figure 4:

**Figure supplement 1.** Activity assay for human DHFR in absence (0 µM) and presence of CD15-3 (17.56–200 µM) carried out at 25°C.

**Figure supplement 2.** Sequence and structural alignment of the binding site for *E. coli* DHFR and human DHFR.

A26T, and L28R (*Palmer et al., 2015*). All mutant *E. coli* strains with the chromosomal *folA* replaced by three drug-resistant variants including P21L, A26T, and L28R exhibit elevated resistance to TMP (*Rodrigues et al., 2016*). In particular, $IC_{50}$ of TMP for *E. coli* strain with chromosomal L28R DHFR is about 80 times higher than of WT (*Table 2*). As shown in *Figure 4A and B*, dose-response curves clearly demonstrated that both CD15 and CD17 inhibit growth of the WT and three single folA mutant *E. coli* strains. The $IC_{50}$ values of CD15 and CD17 can be found in *Table 2*. Importantly, in terms of $IC_{50}$ CD15 is comparable with TMP on the most resistant variant L28R. These results are consistent with the in vitro enzymatic activity assays (*Figure 2C*) showing that both CD15 and CD17 inhibited DHFR activity of L28R variant in vitro stronger than WT, accordingly these compounds inhibited growth of the L28R variant stronger than the WT and the other two mutants.

## Compound optimization

While both CD15 and CD17 exhibited desired biological activity, their $IC_{50}$ for growth inhibition were relatively weak, so we proceeded to optimize both compounds to improve their efficacy. To that end, we searched the ChemDiv database (http://www.chemdiv.com/) for compounds that are similar to CD15 and CD17. The search yielded a total of 12 extra compounds which were subsequently obtained and evaluated for their inhibitory activity against DHFR in vitro and as inhibitors of *E. coli* bacterial growth. The inhibitory $K_i$ values against WT DHFR were in the range of 2–8 µM for the six

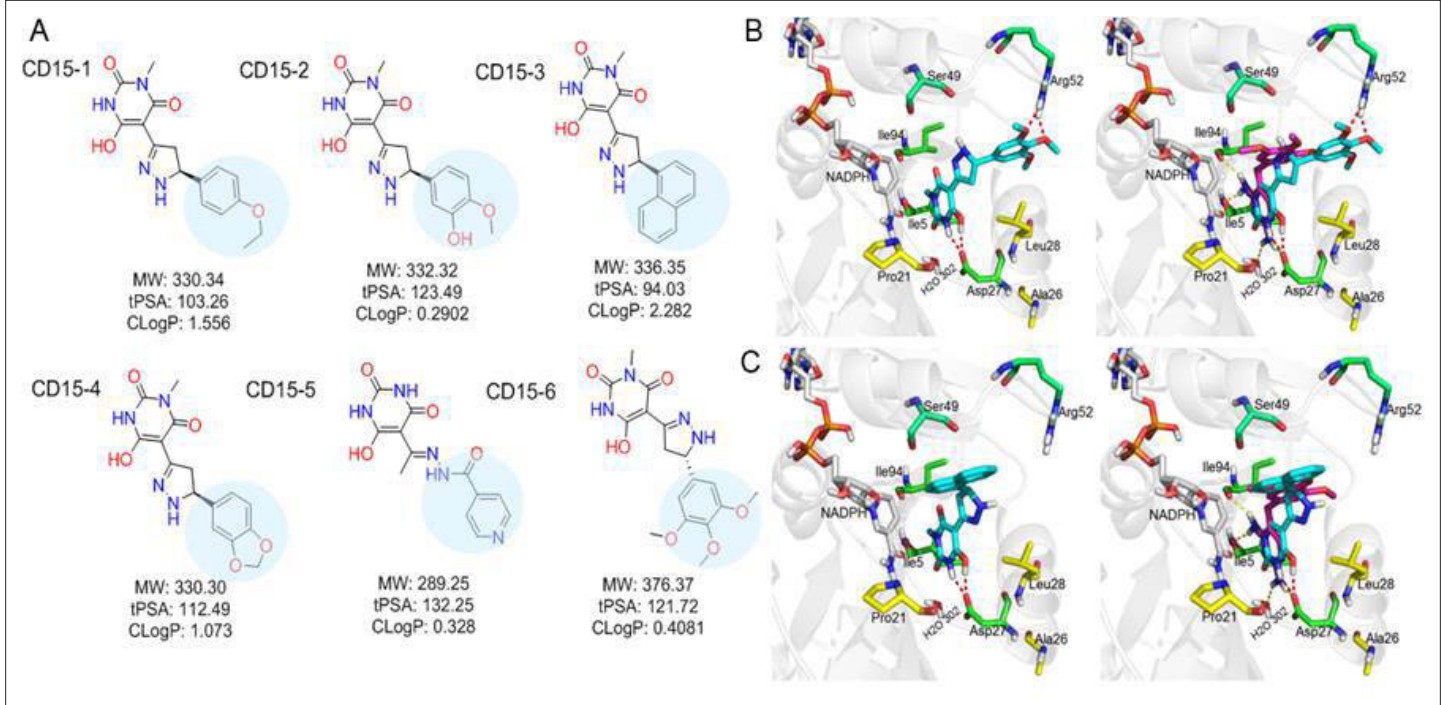

**Figure 5.** Optimization of the compounds of CD15 series (A) Chemical structures of 2nd generation variants of compound CD15 selected for further experimental testing. (**B**) The binding interaction of CD15 with DHFR (left panel) and the alignment of CD15 (cyan stick) with TMP (purple stick) in the binding pocket (right panel). (**C**) The binding interaction of CD15-3 with DHFR (left panel) and the alignment of CD15-3 (cyan stick) with TMP (purple stick) in the binding pocket (right panel).

The online version of this article includes the following figure supplement(s) for figure 5:

**Figure supplement 1.** Growth rate profiles of WT E. coli cells with empty pBAD-plasmid and with WT DHFR and functionally inactive D27F mutant form of DHFR.

compounds from the CD15 series and in the range of 3–30 µM for the six compounds from the CD17 series (*Figure 4C*). Four compounds including CD15-2, CD15-3, CD15-4, and CD15-6 showed better or comparable inhibition for WT and L28R DHFR than CD15 (*Figure 4C*). Next, we evaluated in vivo activity of these compounds. Results showed that the compound CD15-3 with a naphthalene group (*Figure 5A*) instead of the trimethoxybenzene of CD15 (*Figure 2C*) showed marked improvement of efficacy with three to four times lower $IC_{50}$ values compared to that of CD15 (*Figure 4D*). CD15-3 showed about 2.4-fold better efficacy than that of TMP on L28R *E. coli* variant strain (*Figure 4E*). As potential antibiotic leads are supposed to be selective in targeting bacterial protein and not the host protein, we went on to check if CD15-3 interacts and inhibits human DHFR. As shown in *Table 1* no inhibitory activity for CD15-3 was detected against human DHFR. The drop in the fluorescence signal corresponding to the conversion of NADPH and DHFR mediated formation of THF was persistently observed at all concentrations of CD15-3, indicating no inhibition on human DHFR activity (*Figure 4— figure supplement 1*). Structural analysis provided a plausible explanation of selective inhibition of *E. coli* DHFR by CD15-3 and lack of inhibitory activity against human DHFR (*Figure 4—figure supplement 2*). Upon comparison of the crystal structures of *E. coli* DHFR (PDB 1RX3) and human DHFR (PDB 1KMS), we noted there are significant sequence differences between them in the ligand-binding domain (*Figure 4—figure supplement 2*). Different pair of amino acids between *E. coli* DHFR and human DHFR includes Met20: Leu22, Asp27: Glu30, Leu28: Phe31, Trp30: Tyr33, Ile94: Val115. The lack of inhibition on human DHFR for CD15-3 can be attributed to the loss of the pi-pi interaction with Phe31. As for the *E. coli* DHFR, the corresponding amino acid is Leu28, and it does not influence much on the binding with CD15-3.

Improved topological polar surface area (tPSA) and clogP of CD15-3 are likely to be responsible for its superior efficacy of the bacterial growth inhibition on WT and mutant strains than other CD15 series compounds (*Figure 5A*). The $IC_{50}$ values of growth inhibition are listed in *Table 2*. Most of the

CD17 series compounds did not show significantly better efficacy against L28R than the original CD17 (see *Table 2*). To address a possibility that CD17 series is a 'false positive' mostly targeting another protein(s) we turned to pan assay interference compounds (PAINS) filter that seeks compounds which tend to react non-specifically with numerous biological targets simultaneously rather than specifically affecting one desired target (*Baell and Holloway, 2010*). Thus, all 12 CD15 and CD17 series compounds were filtered through the PAINS (http://cbligand.org/PAINS/) (*Baell and Holloway, 2010*). All six CD17 series compounds did not pass the PAINS test and therefore were not considered for further analysis.

The binding mode of the two most promising compounds CD15 and CD15-3 were evaluated using molecular docking (*Figure 1a*, Glide XP mode) with the target *E. coli* DHFR (PDB 1*R*X3). As shown in *Figure 5B and C*, both compounds formed two hydrogen bonds with the key residue Asp27 by the hydroxy group in the 6-hydroxy-3-methylpyrimidine-2,4 (1H,3H)-dione scaffold. The binding modes of CD15 and CD15-3 overlapped perfectly with the binding conformation of TMP, providing rationale for the inhibitory activity of CD15 series. However, in addition to the hydrogen bond formed with Asp27, TMP also forms hydrogen bond with another critical residue Ile94 as well as the conserved water molecule HOH302. Nevertheless, unlike TMP, our hit compounds showed broad inhibitory activity in vitro and in vivo on both the WT and mutant DHFR strains. The broad activity of CD15 compounds can be explained, in part by a dihydro-1H-pyrazole group in the same position as methylene group of TMP. Naphthalene group of CD15-3 extends further in DHFR binding pocket than the corresponding trimethoxybenzene of TMP, potentially resulting in additional hydrophobic interaction with L28R in DHFR which provides structural rationale for strong potency of CD15-3 against resistant L28R variant.

## Target validation in vivo

To confirm DHFR as the intracellular target of CD15-3 we overexpressed DHFR in *E. coli* cells to assess whether it rescues growth inhibition by CD15-3. To that end, we transformed *E. coli* BW27783 with pBAD plasmid (empty plasmid for the control and with folA gene for DHFR expression). BW27783 cells constitutively express arabinose transporters providing rather homogeneous response from the cell pool under arabinose induction (*Bhattacharyya et al., 2016*). Interestingly, controlled expression of folA (encoding DHFR) under pBAD promoter with 0.005% arabinose induction partially rescued growth in a certain range of CD15-3 concentrations (*Figure 6A*). This improvement of growth rate was less pronounced at higher concentration of CD15-3. For control, we used WT cells transformed with empty pBAD plasmids (without folA gene) and observed no effect on growth.

To further probe whether DHFR overexpression rescues CD15-3-induced inhibition of growth by restoring DHFR activity, we overexpressed an inactive variant of DHFR, D27F mutant. To that end, we used the same plasmid system (as was done for WT DHFR) with D27F variant under the pBAD-promoter

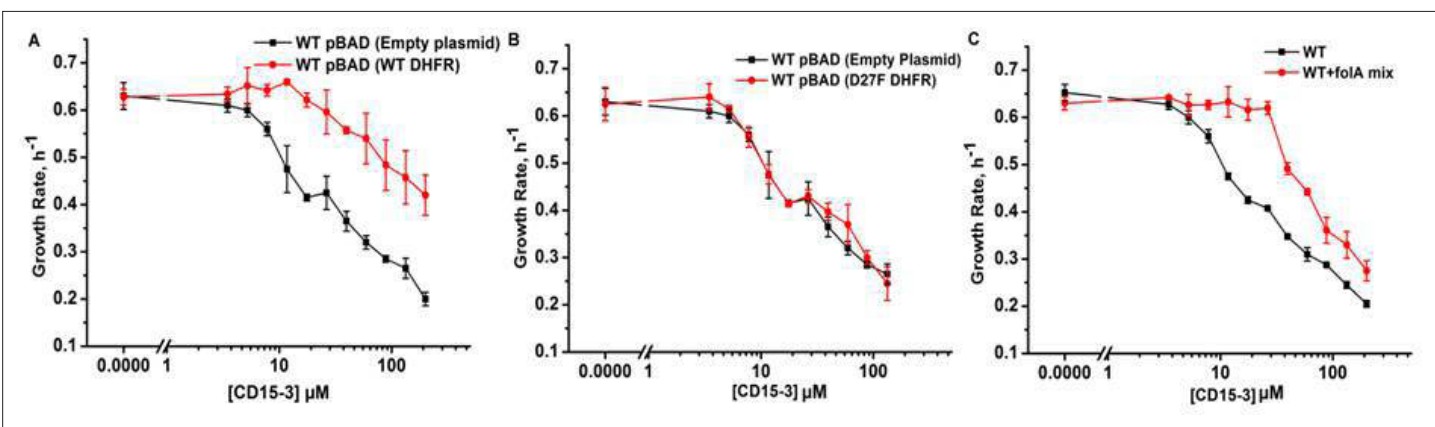

**Figure 6.** Overexpression of functional (WT) DHFR shows partial recovery from CD15-3 induced growth inhibition. (**A**) Overexpression of WT DHFR using pBAD-promoter and at 0.005 % arabinose induction showed improvement in growth rates compared to WT cells with empty pBAD-plasmid (lacking DHFR gene) under conditions of CD15-3 treatment. (**B**) Comparative growth rate profiles of WT cells (with empty pBAD-plasmid) and WT cells overexpressing D27F inactive mutant of DHFR. The growth rate profiles clearly show that D27F mutant of DHFR could not rescue cells from CD15-3-induced growth inhibition. (**C**) Comparative growth rate profiles of WT cells grown in presence of folA mix supplement under conditions of CD15-3 treatment. Cells grown in presence of folA mix metabolic supplementation showed partial rescue in growth under conditions of CD15-3 treatment.

and the expression was induced by 0.005% external arabinose (*Rodrigues and Shakhnovich, 2019*; *Tian et al., 2015*). We found that no growth recovery from CD15-3-induced inhibition upon overexpression of the inactive DHFR variant. The dose response profile was similar to WT cells (with no DHFR overexpression), and same inhibition was observed at all concentrations of CD15-3 (*Figure 6B*). We conclude that loss of DHFR function in vivo due to inhibition by CD15-3 was responsible for partial loss of growth.

This result showed that inhibition of DHFR in vivo was, at least partially, responsible for inhibition of cellular growth induced by CD15-3. As TMP is a known inhibitor of DHFR (WT), we wanted to check if these overexpression plasmid systems behave in a similar way as in the case of CD15-3 inhibition. Overexpression of WT DHFR rescued growth of TMP-treated cells (*Figure 5—figure supplement 1*), while no rescue of growth from TMP-induced inhibition was observed upon overexpression of inactive D27F variant (*Figure 5—figure supplement 1*).

These results indicate that DHFR is an intracellular target for the new compound CD15-3. However, a possibility remains that CD15-3 does not target DHFR exclusively. To understand if CD15-3 is targeting cellular DHFR and thereby disrupting folA pathway, we further performed growth experiments in the presence of supplement folA mix. folA mix which is comprised of purine, thymine, glycine and methionine functions as a metabolic supplement for cells with diminished DHFR function (*Singer et al., 1985*). *E. coli* cells were grown in M9 media supplemented with 'folA mix' in presence and absence of CD15-3. We found that growth of CD15-3-treated cells was partially rescued by folA mix and again the effect was more prominent at relatively lower concentrations of CD15-3 (*Figure 6C*).

At the same time, we observed only partial rescue of CD15-3 inhibited growth by folA mix or DHFR complementation at higher concentrations of CD15-3 suggesting that at high concentrations this compound might inhibit other proteins besides DHFR. In the follow-up publication (*Chowdhury et al., 2021*), we used system-level approaches to discover additional target of CD15-3, besides DHFR.

## CD15-3 largely prevents evolution of resistance in *E. coli*

The primary focus of our approach to design evolution drugs that constrains pathogen's escape routes, is on the search for inhibitors of a single protein target which would be equally effective against WT and resistant variants,. To determine how fast WT *E. coli* can acquire resistance to CD15-3, we evolved three trajectories of *E. coli* under continuous exposure to the drug for over 1 month. We used a previously described automated serial passage protocol (*Rodrigues and Shakhnovich, 2019*) that adjusts the drug concentration in response to increase of growth rate caused by emergent resistant mutations thus maintaining evolutionary pressure to escape antibiotic stressor (see Materials and methods section for details). We found that, at early stages of evolution, the CD15-3 concentration necessary to reduce the growth rate of the culture to 50% (with respect to non-inhibited naïve cells) increased to a value about 2.7 times higher than the $IC_{50}$ of the naïve strain. However, at subsequent time points, the CD15-3 concentration remained constant, indicating that the cells were unable to further develop resistance to CD15-3. We found no mutations in the *folA* locus of the evolved strains upon Sanger sequencing, indicating that the modest increase in resistance to CD15-3 is not associated with target modification. In parallel, we also studied evolution of resistance to TMP in five independent trajectories using the same approach. At the end of the evolutionary experiment, the cells evolved TMP resistance with $IC_{50}$ between 8- and 200-fold higher than original naïve *E. coli* strain (*Figure 7A*). Sanger sequencing of the folA locus revealed point mutations in the folA locus in all trajectories. We found mutations in the DHFR active site, which are associated with resistance to TMP (*Oz et al., 2014*) and also in the promoter region, which are known to increase expression of DHFR (*Tamer et al., 2019*).

To verify the results of evolutionary experiment, the population of cells evolved in presence of CD15-3 was further plated and two colonies were isolated. We measured growth of evolved variant of *E. coli* in M9 media using the same concentration range of CD15-3 as was used for WT cells. Evolved strains exhibited $IC_{50}$ for CD15-3 in the range of 131.5µM to 170.5 µM, about twofold higher compared to $IC_{50}$ for WT (*Figure 7B*). In the other evolution trajectories with TMP selection escape mutants which emerged were W30R, W30C and a culture which was a mixed population of WT cells (with promoter mutation) and cells with L28R folA mutation. $IC_{50}$ values obtained for CD15-3 against TMP escape W30R and W30C were 1.7 and 1.6 times higher compared to naive/WT cells, respectively

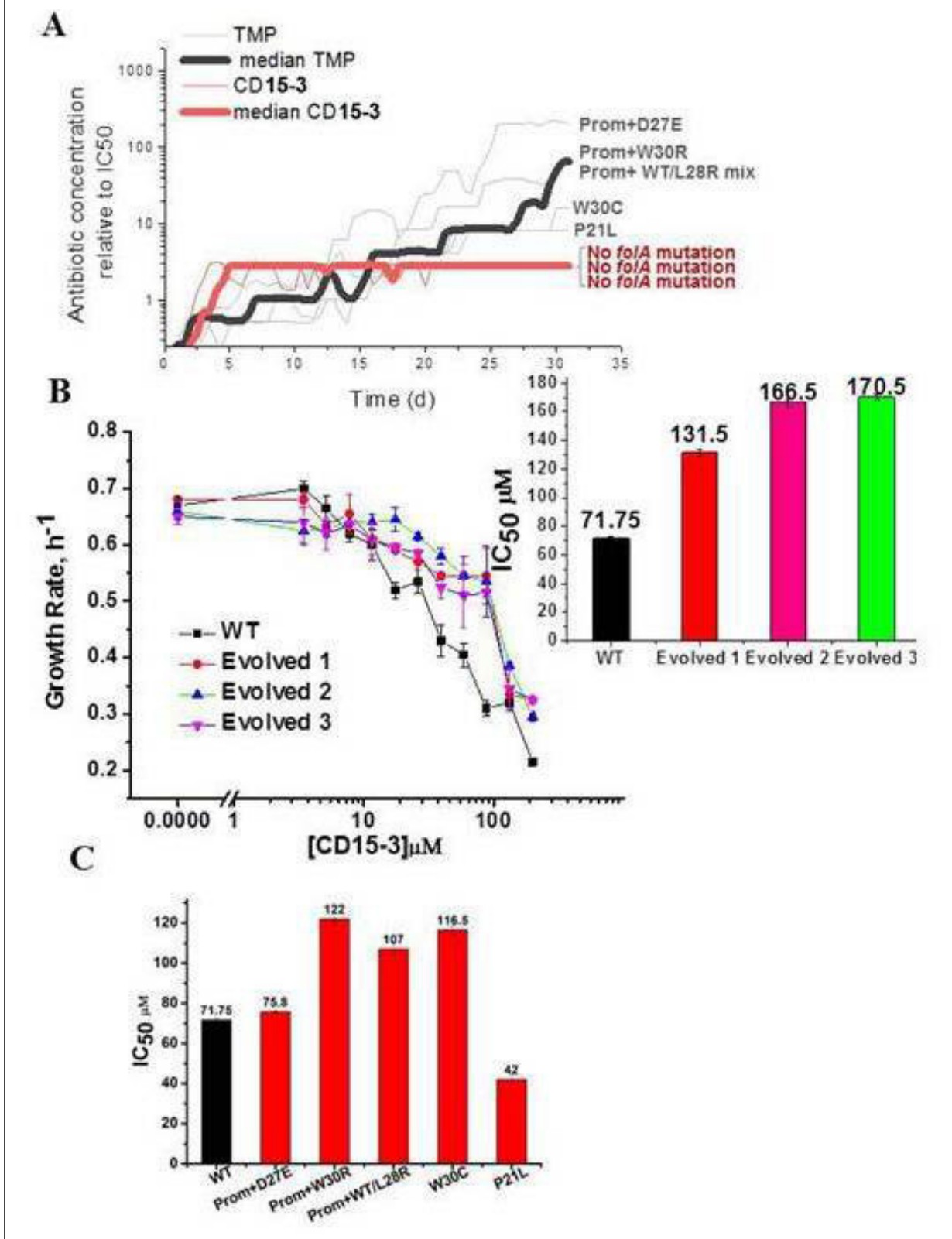

**Figure 7.** Resistance to CD15-3 evolves slowly. (**A**) Gray and red traces show evolution of antibiotic resistance against TMP and CD15-3, respectively. Red (bold) and black (bold) traces shows the median traces of three replicate evolutionary trajectories under selection pressure from CD15-3 and TMP, respectively. Under pressure from TMP cells evolved TMP resistance with IC$_{50}$ between 8- and 200-fold higher than original naïve *E. coli* strain. Cells evolved under CD15-3 treatment showed IC50 values for CD15-3 between 131.5μM and 170.5 μM which is about twofold higher in comparison to IC$_{50}$

*Figure 7 continued on next page*

*Figure 7 continued*

for naive WT. The antibiotic concentrations represented were obtained for a single evolutionary trajectory and are normalized to IC$_{50}$ of naive cells to TMP and CD15-3 (1.3 µM and 71.75 µM, respectively). Cells which evolved under TMP treatment (TMP escape) in individual evolution trajectories showed D27E, W30R, W30C and P21L mutations in the folA locus along with several other mutations outside of folA. One of the TMP trajectories had a mix of WT and L28R population along with promoter mutation. No mutation in the folA locus was observed in the CD15-3 evolved cells. (**B**) Growth rates in WT and evolved strain in a range of CD15-3 concentrations showing weak resistance of evolved strains. Inset shows the differences in IC$_{50}$ values in WT and evolved forms (from three evolution trajectories). (**C**) CD15-3 also inhibits the growth of the TMP escape mutant. D27E escape mutant had similar IC$_{50}$ to CD15-3 as is observed with the naïve/WT. IC$_{50}$ for CD15-3 obtained for W30R, W30C and WT/L28R mix set was found to be less than twofold higher than the WT.

. The TMP trajectory which had a mix of WT cells with promoter mutation and L28R folA mutation had an IC$_{50}$ for CD15-3 which was 1.5 times higher than naive/WT cells.

It is to be noted that CD15-3 was designed to inhibit a subset of TMP escape mutants viz. P21L, L28R, and A26T. Interestingly CD15-3 inhibited the D27E TMP escape mutants along with P21L mutant with IC$_{50}$ close to naïve WT strain (***Figure 7C***). Higher IC$_{50}$ against W30 mutants could potentially mean reduced inhibitory interaction for CD15-3 against these mutants. Further in all these escape mutants, there could be mutations outside folA locus and hence could not be captured by Sangers sequencing of the folA locus. Still IC$_{50}$ for these TMP escape mutants against CD15-3 is less than twofold higher compared to that of WT.

Further, we carried out a complementary plate-based 'one-shot' selection experiment from three independent starter cultures to determine if selection under CD15-3 stress leads to the emergence of colonies with mutations in the folA locus. WT naive cells were grown in M9 media without antibiotic to allow standing variation in the growing culture. At 10$^3$-fold dilution of a culture that has grown for 8 hours, cells were plated on M9 agar plates with either CD15-3 or TMP added. CD15-3 and TMP

**Table 3.** folA locus mutation and IC$_{50}$ values of TMP and CD15-3 resistant colonies which appeared on the TMP and CD15-3 plates.

| | Colony no | Selection | folA mutation | Mean IC$_{50}$ fold increase (w.r.t. WT) | Colony no | Selection | folA mutation | Mean IC$_{50}$ fold increase (w.r.t. WT) |
|---|---|---|---|---|---|---|---|---|
| Culture 1 | 1 | TMP | L28R | 100 | 1 | CD15-3 | no | 1.9 |
| | 2 | TMP | L28R | 110 | 2 | CD15-3 | no | 1.8 |
| | 3 | TMP | M20I | 7 | 3 | CD15-3 | no | 2 |
| | 4 | TMP | no | 10.6 | 4 | CD15-3 | no | 2.3 |
| | 5 | TMP | W30C | 23.3 | 5 | CD15-3 | no | 2.3 |
| | 6 | TMP | no | 14 | 6 | CD15-3 | no | 2.2 |
| Culture 2 | 1 | TMP | no | 12 | 1 | CD15-3 | no | 2.1 |
| | 2 | TMP | no | 10 | 2 | CD15-3 | no | 1.7 |
| | 3 | TMP | L28R | 100 | 3 | CD15-3 | no | 2.3 |
| | 4 | TMP | L28R | 114 | 4 | CD15-3 | no | 2.3 |
| | 5 | TMP | no | 10.4 | 5 | CD15-3 | no | 2 |
| | 6 | TMP | P21L | 2.5 | 6 | CD15-3 | no | 1.9 |
| Culture 3 | 1 | TMP | P21L | 3 | 1 | CD15-3 | no | 2.3 |
| | 2 | TMP | no | 12 | 2 | CD15-3 | no | 2 |
| | 3 | TMP | M20I | 5 | 3 | CD15-3 | no | 1.8 |
| | 4 | TMP | no | 10 | 4 | CD15-3 | no | 1.7 |
| | 5 | TMP | no | 7 | 5 | CD15-3 | no | 1.9 |
| | 6 | TMP | L28R | 110 | 6 | CD15-3 | no | 2.1 |

concentrations were kept three times the respective IC$_{50}$ values (w.r.t. WT i.e. 210 μM CD15-3 and 1.2 μg/μl TMP). A few isolated colonies were found under these conditions. We sequenced folA locus in six isolated colonies from each of CD15-3 and TMP plates and their IC50 values were measured. None of the colonies that grew on CD15-3 plates showed mutation in the folA locus (*Table 3*), while 10 of the 18 colonies isolated from the TMP plate showed mutations in the folA locus. The colonies formed on the CD15-3 plate had IC$_{50}$ values roughly 1.7–2.3 times higher than against WT (*Table 3*). On the other hand the colonies which formed on the TMP plate had much higher IC$_{50}$ values compared to TMP against WT.

In addition to our in vitro evolution study, the plate-based study shows further that CD15-3 escape route does not involve mutation in the folA locus and the CD15-3-resistant forms which emerge upon CD15-3 selection are marginally more resistant compared to the naive/WT cells.

## Whole genome sequencing of the evolved variant

We performed whole genome sequencing of the strains evolved under CD15-3 using two isolated colonies (mentioned as E1 and E2 in *Figure 8A*) obtained from our evolution trajectories keeping naive BW25113 strain as a reference. No mutations at or upstream of the folA locus was found. Therefore, the developed moderate resistance against CD15-3 could not be attributed to target modification.

Further analysis of the sequencing results revealed regions of duplication in the genome of the evolved strain as observed by the double depth-height (*Figure 8A*). Depth or coverage in sequencing outputs refer to the number of unique reads that include a given nucleotide in the sequence. The duplicated segment was found to be a stretch of above 81 KB. In the context of evolution of antibiotic resistance, the relatively frequent occurrence of genome duplications by amplification suggests that evolution of gene dosage can be a faster and more efficient mechanism of adaptation than rare downstream point mutations (*Sandegren and Andersson, 2009*). The gene that confers the limitation is often amplified in this mechanism; however, sometimes increased dosage of an unrelated non-cognate gene can resolve the problem.

Multiple genes belong to the region of genome duplication in the evolved strain (*Figure 8B*) including transporter and efflux pump genes, transposable elements, stress response genes and metabolic genes viz. oxidoreductases, dehydrogenases, kinase regulators etc. The duplicated segment in the genome of the evolved variant contained genes encoding porin proteins, ABC transporter permeases and cation efflux pump genes (cus-genes) (*Supplementary file 2*: Duplicated genome stretch). The CusCFBA system is a HME (heavy metal efflux)-RND system identified in *E. coli*. Resistance-nodulation-division (RND) family transporters refer to a category of bacterial efflux pumps primarily observed in Gram-negative bacteria. They are located in the membrane and actively transport substrates. Cus-efflux system was initially identified for the extrusion of silver (Ag+) and copper (Cu+). They have been found to induce resistance to fosfomycin (Nishino and Yamaguchi, 2001), dinitrophenol, dinitrobenzene, and ethionamide (*Coutinho et al., 2010*). The set of genes constituting cus-system viz. cusCFBA, are all located in the same operon (*Gudipaty et al., 2012*). The system is composed of the RND efflux pump (CusA); the membrane fusion protein, MFP (CusB); and of the outer membrane protein, OMP (CusC). The assembly of these proteins have been reported to be identical to the AcrB, CusA(3):CusB(6):CusC(3) (*Delmar et al., 2013*). In the duplicated segment of the evolved variant, the entire cus-efflux system was found to be present.

We carried out the metabolic characterization of strains evolved in the presence of CD15-3 and naive strains by LC-MS to further investigate the mechanism of resistance to the drug (a detailed analysis is in a subsequent publication) (*Chowdhury et al., 2021*). Interestingly, we found markedly lower abundance of CD15-3 in the evolved strain compared to WT cells suggesting a possible efflux-pump-mediated compound depletion (*Figure 8C*). After 4 hr of CD15-3 treatment, the abundance of the drug was found to be around 10% of the initial abundance. Drug efflux is a key mechanism of resistance in Gram-negative bacteria (*Masi et al., 2017*; *Sandegren and Andersson, 2009*). Pumping out drug compound under conditions of drug treatment is probably the most direct and nonspecific way of combating the toxic effect of a drug. It is interesting to note that we observed higher IC$_{50}$ in the evolved strain compared to the naïve strain for some other antibiotics which we tested. Both the na-ve and CD15-3-evolved cells were treated with TMP and Sulphamethoxazole. Sulphamethoxazole a sulfanilamide, is a structural analog of para-aminobenzoic acid (PABA) and binds to dihydropteroate synthetase. Under both treatment conditions, CD15-3-evolved cells partially escaped the drug

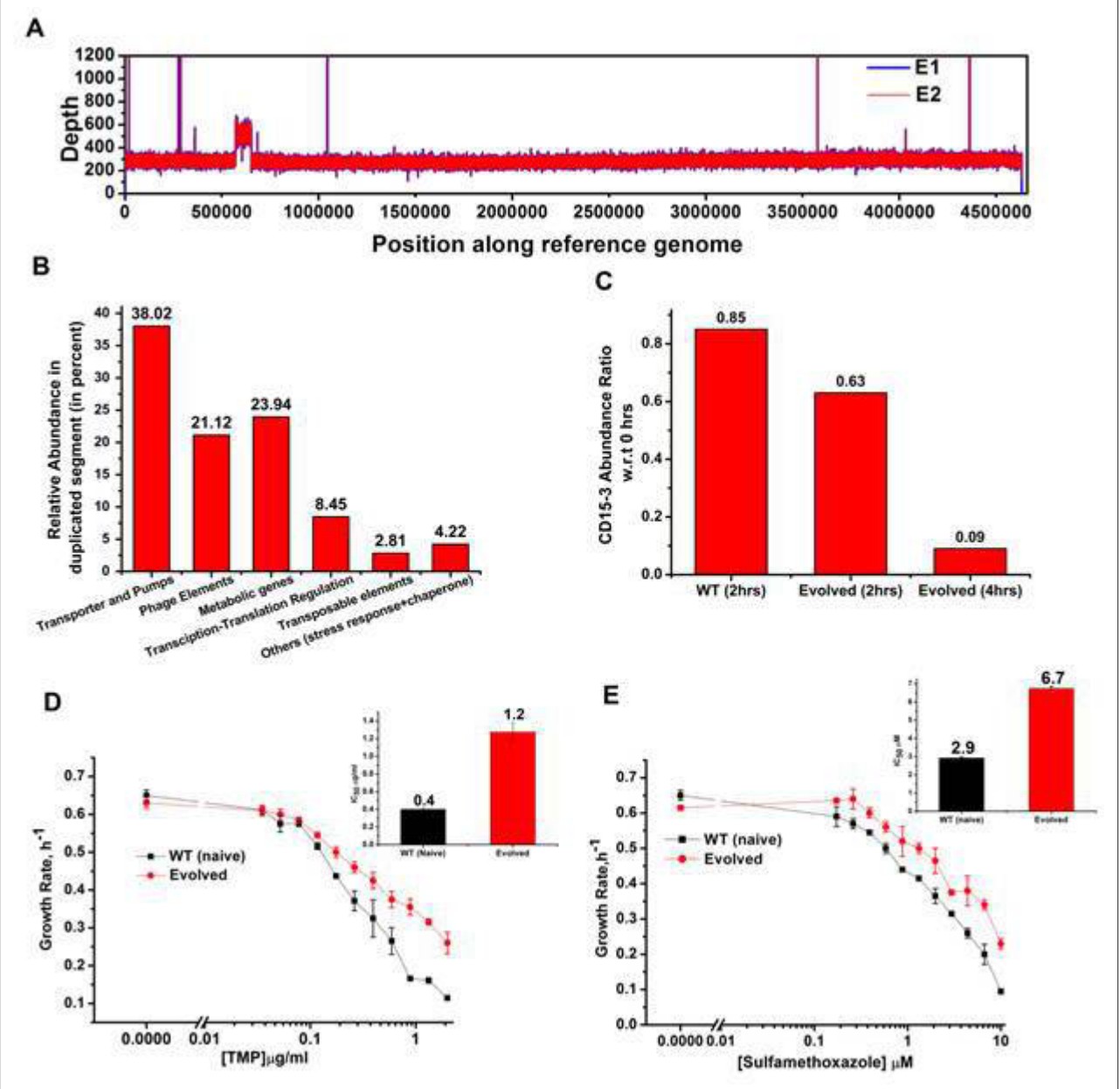

**Figure 8.** Whole genome sequence of evolved variant revealed region of genome duplication. (**A**) Whole genome sequence display of the evolved form on alignment to BW25113 reference genome. The display shows regions of duplication as observed by increased height as per depth axis. (**B**) Bar plot showing the relative abundance of genes which constitutes duplicate segment in the genome of evolved form. (**C**) Bar plot showing the relative intracellular concentration of CD15-3 (with respect to the intracellular concentration in naïve cells at zero hour) in naïve and evolved strains at various time points of treatment. (**D**) CD15-3-evolved cells show cross resistance to other antibiotics. Growth rate profiles of WT (naïve) and CD15-3-evolved cells grown under varying concentrations of trimethoprim (TMP). CD15-3-evolved cells grow better under TMP treatment and have almost three fold higher $IC_{50}$ (shown in inset) compared to WT (naïve). (**E**) Growth rate profiles of WT (naïve) and CD15-3-evolved cells grown in presence of Sulfamethoxazole shows CD15-3-evolved cells grow better compared to WT (naïve) cells (as reflected by the growth rates). CD15-3-evolved cells show somewhat higher $IC_{50}$ compared to WT (naïve) when grown in presence of Sulfamethoxazole (shown in in inset).

inhibition and showed about threefold higher $IC_{50}$ for TMP and Sulphamethoxazole (*Figure 8D and E*). These results show that the efflux mediated drug resistance in the evolved strain is non-specific. It demonstrates a potential strategy for antibiotic cross resistance and helps bacteria to escape inhibitory actions of CD15-3 and other antibacterial compounds with completely different protein targets. In the same vein, we note that the efflux pump mechanism shows only moderate increase of $IC_{50}$ for

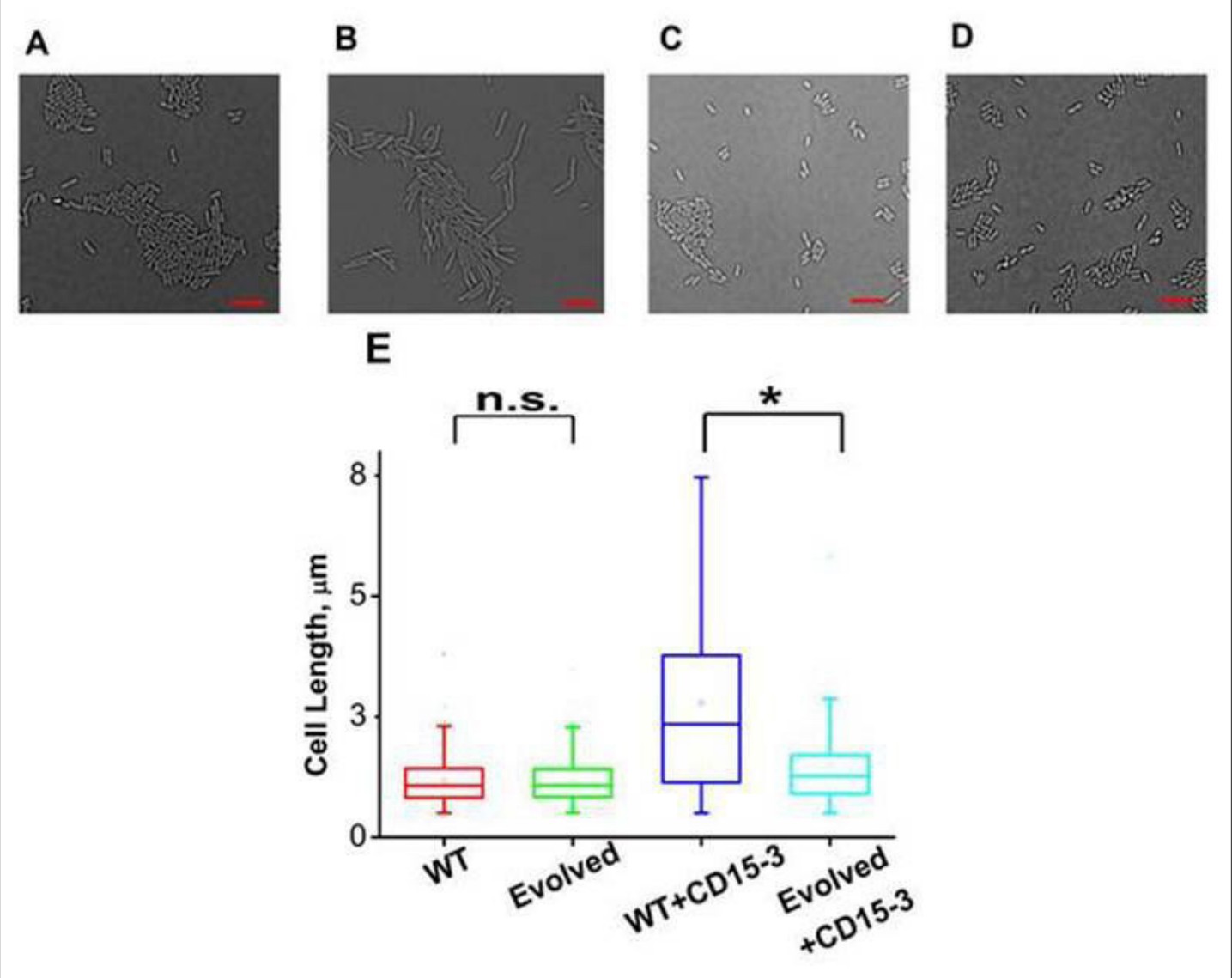

**Figure 9.** CD15-3 treatment leads to stress-induced morphological changes in WT *E. coli* cells. DIC image of WT cells under (**A**) control (no CD15-3 treatment) and (**B**) treated (CD15-3 treatment) conditions. (**C**) DIC image of evolved *E. coli* cells under control (no CD15-3 treatment) and (**D**) CD15-3-treated condition. (**E**) Distribution of cell lengths as derived from DIC imaging of WT and evolved cells under control and CD15-3 treatment conditions. Scale bar corresponds to a cell length of 2 μm. Untreated WT (naive) and evolved cells had comparable cell lengths with median cell lengths of 1.07 μm. (n.s. indicates the distribution of cell lengths were not significantly different, Mann-Whitney test, p-value = 0.961). Both WT and evolved cells were subjected to CD15-3 treatment at $IC_{50}$ concentrations. WT-treated cells were observed to have a filamentous morphology and the median cell length (2.34 μm.) double of that of the untreated WT set. Evolved cells after CD15-3 treatment had a median cell length of 1.26 μm which is slightly higher than that of untreated set. But the cell size distribution of the evolved cells showed much less change after CD15-3 treatment compared to that observed for the WT (* indicates the distribution of cell lengths were significantly different, Mann-Whitney test, p-value < 0.001).

a variety of antibiotics in contrast to about 200-fold increase in several strains evolved under TMP (*Figure 7*).

## Morphological changes induced by CD15-3 treatment

As cellular filamentation and concomitant morphological changes are one of the visible hallmarks of stress responses to inhibition of proteins on the folate pathway of which DHFR is a member,(*Ahmad et al., 1998*; *Justice et al., 2008*; *Sangurdekar et al., 2011*; *Zaritsky et al., 2006*) we proceeded to image *E. coli* cells in absence of CD15-3 (control) and with added CD15-3. Experiments were performed at 42°C. Cells were grown for 4 hours before image acquisition. DIC imaging results of the

CD15-3-treated WT cells (*Figure 9B*) showed filamentation as compared to the cells grown in absence of CD15-3 (*Figure 9A*). Treated cells showed a wide distribution of cell length with a median length which was more than double than that of the untreated sets (*Figure 9E*).

Similar imaging experiment was also carried out for the CD15-3-resistant variant of *E. coli* cells obtained in our evolution experiment. These cells (*Figure 9C*) had similar median cell lengths as WT (naive) cells before CD15-3 treatment. Upon CD15-3 treatment, these evolved cells showed slightly higher median cell length, although no visible filamentation was observed (*Figure 9D*). As efflux genes comprise 38 % of the duplicated genome segment (*Supplementary file 2*: Duplicated genome stretch) and CD15-3 abundance after 4 hours of treatment was markedly lower (compared to naive WT) an efflux-mediated drug resistance mechanism could potentially explain as to why evolved cells did not show any CD15-3-induced stressed morphology. The slight increase in median cell length in evolved cells upon treatment could be attributed to the fact that the transporter and efflux-pump-mediated resistance is a rather generic way to escape drug action and is limited in its ability to completely evade drug action.

## Discussion

Antibiotic resistance has emerged as a critical threat in the treatment of infectious diseases. This challenge is primarily an evolutionary problem which nails down to factors like antibiotic induced selection, the population structure of the evolving microbes and their genetics of adaptation (*Roemhild and Schulenburg, 2019*). Horizontal gene transfer – another critical mechanism conferring antibiotic resistance (*Dzidic and Bedeković, 2003*) in bacterial population – presents further complications. Traditional antibiotic design protocols and development pipelines do not essentially consider this high intrinsic potential of bacterial adaptation in antibiotic stressed growth conditions and hence do not properly address the problems associated with drug resistance. This problem has become more acute with the emergence of multidrug resistant bacteria which can potentially escape the actions of a wide array of antibiotic formulations and have a complex landscape of evolutionary adaptation.

To tackle these problems, complex treatment strategies for infectious diseases like combination-drug therapies or sequential drug therapies have been developed (*Baym et al., 2016*; *Vestergaard et al., 2016*). These strategies are considered to be superior to classical monotherapies as instead of one protein target they aim at multiple molecular targets. It is interesting to note that natural product antibiotics like penicillin and other beta lactam drugs are inherently multi-target antibiotics (*Tyers and Wright, 2019*). Same applies to tetracyclin and aminoglycosides which targets bacterial ribosome and in turn interact with multiple ribosomal proteins (*Forge and Schacht, 2000*). All these prompted interest in targeting multiple molecular targets with an aim to suppress the emergence of antibiotic resistance by target modification (*Silver, 2007*).

However, in many cases, these strategies enjoyed only limited success in constraining the emergence of evolutionary escape variants and rare multidrug variants (*Hegreness et al., 2008*). Furthermore, none of these strategies act proactively by suppressing future evolutionary escape variants and remain more as a monotherapy in sequence or a combination therapy targeting WT variants.

Evolution drugs were envisioned as chemical formulations in the potential anti-evolution strategy which would prevent or delay evolution of drug resistance. Recent studies highlighted the bacterial evolutionary landscape under antibiotic induced selection pressure and the potential drug escape routes (*Cao et al., 2017*; *Palmer et al., 2015*; *Ogbunugafor et al., 2016*; *Rodrigues et al., 2016*). In a parallel development, the evolutionary dynamics of viral escape from drug or antibody stress was studied in silico and in vivo (*Doud et al., 2018*; *Gaebler et al., 2021*; *Louie et al., 2018*; *Marchi et al., 2019*; *Rotem et al., 2018*) This knowledge is critical to design better strategies to block the emergence of drug or antibody resistant phenotypes of pathogens.

One of the best-studied examples is evolving resistance to inhibitors of DHFR. Rapid clinical resistance to available antifolates like TMP emerges merely after three rounds of directed evolution and sequential fixation of mutations through ordered pathways has been shown to contribute to the evolutionary paths for antibiotic resistance (*Tamer et al., 2019*; *Toprak et al., 2012*). Even though several classes of small molecules have been investigated for their potential antifolate activity, the rapid emergence of resistance by readily accessible mutational pathways in the folA gene pose an immense challenge (*Toprak et al., 2012*). Several studies have shown that *E. coli* cells evolved in the presence of trimethoprim, a DHFR inhibitor, can become virtually insensitive to the drug upon successive

accumulation of mutations in target gene. It is thus urgent to develop new tools for the systematic identification of novel scaffolds and discovery of inhibitors that interact simultaneously with both WT DHFR and antifolate resistant DHFR mutants. Importantly, the development of a biophysics-based quantitative model for the fitness landscape of DHFR (*Rodrigues et al., 2016*; *Rodrigues and Shakhnovich, 2019*) is a valuable tool to identify the target-resistant mutants that are the best candidates for drug design based on biophysical parameters of mutant proteins. The current work presents a design protocol and a multi-tool drug development pipeline which is motivated by these evolutionary understandings and presents a plausible strategy which aims at inhibiting both WT and evolutionary escape variants.

By an efficient virtual screening protocol to scan large databases, two hits of novel scaffolds were identified, which showed inhibitory activity against both WT and resistant mutant forms of DHFR. Unlike conventional approaches primarily focused on chemical synthesis of derivatives based on known inhibitor scaffolds (*Francesconi et al., 2018*; *Hopper et al., 2019*; *Lam et al., 2014*; *Reeve et al., 2016*), we deployed a multi-tier approach to come up with chemically novel inhibitors. To further improve DHFR inhibitory activity of the two hits, a rapid round of compound optimization was conducted through a structural similarity search. This approach turned out to be effective as we tested only 12 candidate compounds and found CD15-3 with improved antimicrobial activity against both the WT and trimethoprim-resistant DHFR mutant *E. coli* strains. Among trimethoprim-resistant mutations, P21L, A26T, L28R, and their combinations recurrently appeared in two out of five independent evolution experiments, and their order of fixation in both cases was similar (*Toprak et al., 2012*). Fitness landscape of TMP resistance showed that the L28R variant is most resistant among all three single mutants (*Rodrigues et al., 2016*). Introduction of another mutation on the L28R background, either P21L or A26T, does not change $IC_{50}$ significantly (*Rodrigues et al., 2016*; *Toprak et al., 2012*). Thermal denaturation experiments demonstrate that L28R mutation is stabilizing showing an increase in Tm of 6°C above WT DHFR, in contrast to the destabilizing mutations P21L and A26T (*Rodrigues et al., 2016*). In addition, the L28R mutation cancels out the destabilization brought by P21L and A26T, restoring the Tm of the double and triple mutants to WT values. The L28R mutation also shows improved compactness, inferred from its reduced bis-ANS binding (*Rodrigues et al., 2016*). Thus, among the evolved mutant *E. coli* strains, L28R is one of the most frequently encountered resistance mutations in folA gene which render traditional anti-folate drugs like TMP clinically ineffective (*Manna et al., 2021*; *Rodrigues et al., 2016*; *Toprak et al., 2012*). Effectively L28R serves as a stability 'reservoir' providing a gateway to multiple evolutionary trajectories leading to resistance. The discovered DHFR inhibitor CD15-3 shows about threefold better efficacy than that of TMP on L28R *E. coli* variant strain, indicating its potential in combating resistance conferred by gateway point mutations of DHFR.

Our in vivo DHFR over-expression experiment showed partial recovery from the CD15-3-induced growth inhibition, thereby validating DHFR as a target for the intracellular inhibition by CD15-3. That DHFR is being targeted by CD15-3 is further supported by our imaging results showing antibiotic-stress-induced filamentation in the WT cells, a clear sign indicating that the folate pathway is indeed impacted by the drug (*Bhattacharyya et al., 2021*). However, the *partial* recovery upon DHFR over-expression strongly suggests the presence of at least one additional protein target of CD15-3. As CD15-3 was designed to allow interaction with both WT and 'modified' active site pockets in the mutant forms of DHFR, this might make CD15-3 active against other enzymes that bind structurally similar substrates, most likely of the folate pathway. In a follow-up work, we present integrative metabolomic analysis that allowed us to identify additional targets of CD15-3 (*Chowdhury et al., 2021*).

In order to assess the validity of the strategy to design compounds that aims to constraint evolutionary escape routes we carried out 30-day long evolution experiments to evolve *E. coli* under CD15-3-induced stress. Unlike TMP resistance which emerged quickly and resulted in strains having 8- to 200-fold higher $IC_{50}$, under CD15-3 *E. coli* evolved $IC_{50}$ which was only 2.7 times greater than WT. Further our whole genome sequence analysis revealed that there was no mutation in the target folA gene. Rather, we found that partial genome duplication of the region containing efflux pumps was responsible for evolution of modest resistance against CD15-3. The importance of efflux pump genes and transporters as first line of defence against antibiotic stressors was shown in studies where deletion of such genes leads to inferior mutational paths and escape strategies (*Lukačišinová et al., 2020*). The generic resistance phenotype observed in CD15-3-evolved cells could be potentially attributed to the multivalent nature of CD15-3. TMP specifically interacts with DHFR and hence mutation in the

target loci (folA) provides an optimal solution to escape the TMP stress. As CD15-3 potentially inter-acts with at least one more target besides DHFR, point mutations in one of the target proteins do not provide optimal solutions. Hence resistance to CD15-3 is less likely to evolve through mutations in folA. While point mutations at multiple target loci can complicate the evolutionary fitness landscape, efflux-mediated resistance on the other hand provides a generic and rapid rescue strategy bypassing the need of multiple point mutations. However, this strategy apparently has its drawback as overex-pression of additional pumps can incur fitness cost and lead to only modest two- to threefold increase in $IC_{50}$.

Bacterial fitness/adaptation landscape under conditions of antibiotic induced stress could be complicated with multiple escape strategies. The method and the plausible solutions presented here aim to make common antifolate evolutionary escape route less accessible through missense mutations in the folA locus. Indeed CD15-3 was effective in constraining the escape routes involving target modifications by inhibiting the most potent DHFR escape mutants along with the WT form. Surprisingly, CD15-3 was equally effective in inhibiting growth of the TMP escape mutants which were not explicitly modeled at the computational design stage of this project such as D27E. CD15-3 also showed modest inhibition of W30R and W30C mutants emerged under TMP selection with $IC_{50}$ values less than twofold higher than the WT. It is interesting to note that the CD15-3 inhibitory activity extends beyond the DHFR variants that were initially selected as targets for structure-based design and that "unplanned" D27E and W30 mutants get inhibited with WT like efficacy.

Exploration of long-term evolutionary mechanism of potential escape from CD15-3 can help us understand the complexity of fitness landscape for this novel prospective antibiotic lead. This can further help in refining the drug development protocol which can also address the issue of efflux-mediated evolutionary escape routes. A plausible futuristic anti-evolution strategy could be using combination of potential evolution drug formulations like CD15-3 and adjuvants which could block the pumps, thereby potentially also obstructing the generic escape routes. Further this anti-evolution strategy in principle would be interesting to explore in the context of the currently known combination-drug therapy protocols extending its scope relevance and application. If CD15-3 has additional non-DHFR cellular target, it could serve as a monotherapy-mimic of a combination drug cocktail whose design is guided by evolutionary understanding based on recent insights into a short-term bacterial adaptation. We suggest that the currently popular drug development protocols could be improved to proactively account, at least partly, for bacterial adaptation.

As a caveat we note that novel compounds discovered here, while promising as candidates for new class of anti-evolution antibiotics, represents only initial leads in the drug development pipeline. A detailed characterization of pharmacokinetics of the prospective compound using the standard approaches adopted in pharmaceutical industry needs to be carried out to establish its plausibility as drug candidate. However, our approach that integrates computational and in vitro experimental components in a mutual feedback loop allows considerable power and flexibility in optimization of emerging compounds potentially significantly shortening the development cycle.

To summarize, we believe that the presented comprehensive multiscale-multitool approach can be an attractive starting point for further optimization and development of evolution drugs and adopting a broad strategy to combat evolution of drug resistance.

## Materials and methods
### Preparation of compound databases

Two commercially available databases ChemDiv (http://www.chemdiv.com/, about 1.32 million compounds) and Life Chemicals (http://www.lifechemicals.com/, about 0.49 million compounds) were selected for virtual screening. Both databases contain a large amount of diverse structures which are useful as potential hits for drug development and providing integrative drug discovery strategies for pharmaceutical and biotech companies (*Zhang et al., 2015*). In addition, the known *E. coli*. DHFR inhibitors (183 compounds) with $IC_{50}$ values ranging from 1 nM to 100 µM were downloaded from BindingDB (https://www.bindingdb.org/bind/index.jsp) *Liu et al., 2006* used as a reference database. All compounds from the two databases were added hydrogens and then optimized by a user-defined protocol in Pipeline Pilot (*Warr, 2012*). A couple of physicochemical properties including molecular weight, AlogP, number of hydrogen donors (HBD) and acceptors (HBA), number of rings, aromatic

rings, rotatable bonds and molecular fractional polar surface area (Molecular_FPSA) were calculated in Pipeline Pilot (**Warr, 2012**) using the Calculate Molecular Properties protocol. All the compounds were further prepared by the LigPrep module in Maestro 10.2 in Schrodinger (**Steffan and Kuhlen, 2001**) to ensure they were appropriate for molecular docking.

## Molecular docking

All *E. coli* DHFR crystal structures (issued before 09-10-2016) were downloaded from Protein Data Bank (PDB) (**Berman et al., 2007**) and classified into three categories. Among them four representatives including PDB 1RX3 (M20 loop closed), 1RA3 (M20 loop open), 1RC4 and 5CCC (M20 loop occluded) were selected for molecular docking and molecular dynamics studies. All these protein structures were prepared by the Protein preparation wizard in Maestro 10.2 of Schrodinger (**Madhavi Sastry et al., 2013**) to add hydrogens and missing atoms and to minimize the energy. All water molecules in the original crystal structures were removed except the $302^{th}$ $H_2O$ which is located in the active site and forms a key hydrogen bond with the cognate ligand. Hydrogen atoms and partial charges were added to the protein, and then a restrained minimization of only the hydrogens was conducted. A grid box centered on the cognate ligand was generated by the Receptor Grid Generation module in Schrodinger. To soften the potential for nonpolar parts of the receptors, one can scale the van der Waals radii (scaling factor set to 1.0) of the receptors with partial atomic charge less than the specified cutoff (set to 0.25 Å). Due to its excellent performance through a self-docking analysis (**Zhang et al., 2017**), the Glide module in Schrodinger was selected for molecular simulations. All three precision modes including the high-throughput virtual screening (HTVS), the standard precision (SP) and the extra precision (XP) were utilized sequentially according to their speed and accuracy. For each docking run, the 10 best poses of each ligand were minimized by a post docking program with the best pose saved for further analysis and the root mean standard deviation (RMSD) between the output and input structures were calculated.

## Molecular dynamics

Protein-compound complexes resulted from molecular docking were employed for molecular dynamics simulation. Proteins were prepared using the Protein Preparation Wizard (**Madhavi Sastry et al., 2013**) tool of Maestro in Schrodinger by adding missing atoms and hydrogens and minimizing the energy. Ligands were preprocessed by assigning the correct protonation state using Chimera 1.9 (**Pettersen et al., 2004**). Then, they were optimized by an implicit water model under the M06-2X/6–31+ G** level of theory and their RESP charges were computed at the HF/6–31 G* level of theory using Gaussian 09 (**Chai and Head-Gordon, 2009**). Finally, their ligand topologies were generated by acpype which is an interface to antechamber (**Wang et al., 2006**). The TIP3P water model (**Jorgensen, 1981**) and AMBER03 force field (**Chillemi et al., 2010**) were used for the simulation using GPU processors with GROMACS, version 5.0.2 (**Spoel et al., 2005**). Firstly, the system was solvated in a water filled rhombic dodecahedron box with at least 12 Å distance between the complex and box edges. Then, the charges of system were neutralized by adding counter ions (Na+ or Cl-) by the genion tool in Gromacs. The system was then relaxed through an energy minimization process with steepest descent algorithm first, followed by a conjugate gradient algorithm. The energy of the system was minimized until the maximum force reached 5.0 kJ mol⁻¹ nm⁻¹. After a primary constraint NVT simulation of 200 ps with protein atoms being fixed at 100 K, an NVT simulation of 400 ps without restraints was performed with simulated annealing at the temperature going from 100 to 300 K linearly. Then we performed an NPT simulation of 500 ps to equilibrate the pressure. Eventually, a production MD was conducted for 10 ns at 300 K twice. Bond lengths were constrained using LINCS algorithm during the simulation (**Hess, 2008**). It is commonly accepted that nanosecond MD simulations are reliable to investigate local conformational changes (**Yeggoni et al., 2014**). Thus, the last 2 ns of each 10 ns production simulation were extracted at every 10 ps interval (400 snapshots in total) for calculating the molecular mechanics/Poisson-Boltzmann surface area binding energy (MM-PBSA) by the AMBER12 package (**Miller et al., 2012**). Average energy for 400 snapshots was saved for binding free energy calculation using MMPBSA (**Cheron and Shakhnovich, 2017**). More details about this protocol are described elsewhere (**Cheron and Shakhnovich, 2017**).

## Crystal structure selection

Through visual inspection of M20 loop conformation for the 61 crystal structures of DHFR (before 2016-9-26), crystal structures of three categories of M20 loops were obtained. From *Figure 1—figure supplement 1*, a total of 38 closed, 11 open, 12 occluded crystal structures of *E. coli* DHFR as well as the comparison of them are shown. The NADPH/NADP + as well as the substrates (most of them are MTX) are also shown. It can be seen that M20 loop constitutes part of the substrate binding site, so it will influence the binding of the substrates and its inhibitors. It has shown that the closed conformation is adopted when the substrate and cofactor are both bound (*Agarwal et al., 2002*), where the M20 loop is packed against the nicotinamide ring of the cofactor. This is the only conformation where substrate and cofactor are arranged favorably for reaction and the following state of the catalytic cycle (*Agarwal et al., 2002*) and also the only conformation that can be found in DHFR from all other species, regardless of the crystal packing and ligands bound positions in the binding site (*Sawaya and Kraut, 1997*). Observed in the product complexes, the occluded M20 loop is an unproductive conformation in which the central part of the M20 loop forms a helix that blocks access to the binding site for the nicotinamide moiety of the cofactor. Thus, the nicotinamide of the cofactor is forced out into solvent, making it unresolved in the crystal structures (*Sawaya and Kraut, 1997*). The open loop is a conformational intermediate between the extremes of the closed and occluded loops. The M20 loop dynamics plays a significant role in ligand binding and catalysis. It has become an area of great interest due to the persuasive evidence of conformational change in the loop during the catalytic cycle and its interaction with the substrate and cofactor (*Sawaya and Kraut, 1997*). However, it is still unknown which M20 loop conformation is a binding mode for inhibitors of *E. coli* DHFR. Whether binding single conformation or multiple conformations of the M20 loop leads to stronger binding is also not known. Thus, it is important to determine the target crystal structures that is most predictive for virtual screening investigation. Considering the conformation M20 loop, crystal structure resolution, as well as completeness of cognate ligands such as MTX or DDF and NADPH, we selected four M20 loop conformations (Closed: 1*R*X3, Open: 1RA3, Occluded: 1RC4 and 5CCC, *Figure 1—figure supplement 1*) for the docking verification. Two occluded PDBs were selected because their M20 loop conformations did not overlap well. The feasibility of each M20 loop conformation was assessed by docking the known 183 *E. coli* DHFR inhibitors with IC$_{50}$ values ranging from 1 nM to 100 μM to the four representative PDB structures. As shown in *Figure 1—figure supplement 2* and *Table 4*, for docking score less than –10, the cumulative numbers of selected inhibitors were 22 for the closed M20 1*R*X3 – greater than for the open M20 1RA3 (10), and occluded M20 1RC4 (1) and 5CCC (13). In particular, when the cutoff of docking score was set to –9, using closed M20 1*R*X3 as target resulted in correct prediction of as many as 62 inhibitors compared to the open M20 1RA3 (25), and occluded M20 1RC4 (4) and 5CCC (32). At the higher cutoffs, open M20 1RA3 and occluded M20 5CCC showed comparable trends with that of closed M20 1*R*X3. However, even the M20 loop of 1RC4 and 5CCC both belong to the occluded conformation, their performance in docking of known ligands was strikingly different. It appears that 1RC4 performed much worse than 5CCC and it performed the worst

**Table 4.** Number of known inhibitors binding DHFR with different M20 loops through molecular docking.

| XP docking score threshold | All inhibitors (IC$_{50}$: 1 nM~240 μM) | | | | Inhibitors (IC$_{50}$ <1 μM) | | | |
|---|---|---|---|---|---|---|---|---|
| | Closed (1*R*X3) | Open (1RA3) | Occluded (1RC4) | Occluded (5CCC) | Closed (1*R*X3) | Open (1RA3) | Occluded (1RC4) | Occluded (5CCC) |
| < –10 | 22 | 10 | 1 | 13 | 14 | 4 | 0 | 4 |
| < –9 | 62 | 25 | 4 | 32 | 42 | 13 | 2 | 13 |
| < –8 | 77 | 77 | 12 | 73 | 48 | 50 | 4 | 35 |
| < –7 | 107 | 93 | 21 | 114 | 51 | 53 | 8 | 51 |
| < –6 | 148 | 136 | 53 | 167 | 58 | 58 | 22 | 68 |
| < –5 | 174 | 168 | 148 | 181 | 69 | 67 | 53 | 72 |
| < –4 | 181 | 182 | 182 | 183 | 72 | 72 | 71 | 72 |
| < 0 | 183 | 183 | 183 | 183 | 72 | 72 | 72 | 72 |

compared to other target conformations. A possible reason could be that M20 loop in 1RC4 structure is closer to the substrate, making the binding pocket smaller to accommodate relatively large inhibitors. Further, we focused on the inhibitors with $IC_{50}$ values less than 1 μM. As seen from *Figure 1— figure supplement 2* and 3, using the closed M20 1RX3 predicted about 58 % (42) out of the total of 72 the inhibitors, while the open M20 1RA3 only predicted 18 % (13), the occluded 1RC4 got 3 % (2) and the occluded 5CCC predicted about 18 % (13). Altogether these results demonstrate that the closed M20 1*RX*3 is more representative as a target for molecular docking of *E. coli* DHFR inhibitors.

## Construction of binding affinity prediction model

Earlier, the MD simulation of protein-ligand complexes followed by MM/PBSA assessment of binding affinity were applied in our group to the BACE protease (*Cheron and Shakhnovich, 2017*). Here, we use a similar protocol for *E. coli* DHFR. Eight known *E. coli* DHFR inhibitors (*Figure 1—figure supplement 4*) including MTX and TMP as well as other six compounds from *Carroll et al., 2012* with known Kd values were chosen for the construction of binding free energy prediction. The Pearson correlation coefficient (R) between the predicted and experimental binding free energies (ΔG = RTln(Kd), where R is the gas constant and T = 293.15 K) was used to evaluate the accuracy of the protocol developed in *Cheron and Shakhnovich, 2017*. TMP and Cmpd two as well as Cmpd 6 (*Figure 1—figure supplement 4*) do not have crystal structures, their complexes were from molecular docking (TMP was docked to PDB 1*RX*3 and Cmpd 2 and 6 were docked to 3QYL). We used the available crystal structures of TMP complexed with Staphylococcus aureus DHFR (PDB: 2W9G, sequence identity with *E. coli* DHFR: 55/162 (34.0%) and sequence similarity: 92/162 (56.8%)) as a reference and found that the conformation of TMP docked with *E. coli* DHFR is only about 0.4 Å different from crystal structure with *S. aureus* DHFR (*Figure 1—figure supplement 5*), suggesting the accuracy of docked conformation for TMP. We first conducted two replicates of 10 ns simulation and then extended simulation length to 20 replicates. Two replicates of 10 ns simulation provided a quite strong correlation (*R* = 0.942) between the computational and experimental binding free energies (*Figure 1B*) essentially indistinguishable from longer simulation comprised of 20 replicates of 10 ns runs in *Figure 1—figure supplement 6*. Thus, in subsequent simulations, a protocol consisting of two replicates of 10 ns was adopted. It can also be observed that the binding free energies followed a normal distribution (*Figure 1—figure supplement 6*) in both the short and long simulations, which indicates that the use of mean value to represent the general binding free energy is reasonable. An evolutionary study (*Oz et al., 2014*; *Toprak et al., 2012*) of TMP resistance found that three key resistance mutations P21L, A26T, L28R, and their combinations constitute a set that recurrently occurred in two out of five independent evolution experiments, and their order of fixation in both cases was similar. Although mechanisms leading to TMP resistance could be extremely complex and diverse, these sets of point mutations represent a simplistic model of in vitro evolution. These point mutations are located within the binding pocket of dihydrofolate substrate within a small region of eight residues in the DHFR protein that constitutes a flexible Met-20 loop (residues 9–24) and an α-helix (residues 25–35). Thus, the correlation coefficients between the computational and experimental binding free energies for TMP for the three mutants as well as their double and triple combinations were calculated. In addition, linear regression equation models to predict binding free energy for TMP against mutant DHFR originated from *Listeria grayi* and *Chlamydia muridarum* were also included. As shown in *Figure 1—figure supplement 7*, from two replicates of 10 ns simulation, the correlation coefficient (R) for *L. grayi* and *C. muridarum* DHFR were 0.895 and 0.839, respectively. The reason why the prediction for *L. grayi* and *C. muridarum* is somewhat inferior to *E. coli* DHFR (R = 0.953 from *Figure 1C* in the manuscript) may be that the protein-ligand complexes used for MD simulation are derived from molecular modeling rather than from crystal structures. Still, the values of binding free energy follow a normal distribution. Those models were used for scoring of virtual screening hits later.

## Selection of virtual screening hits

Structure-based virtual screening (SBVS) can quickly select compounds with reasonable binding patterns and higher predicted scores from a large number of compounds. According to the workflow (*Figure 1A* in the manuscript), a total of about 1.8 million compounds from ChemDiv and Life Chemicals were screened for compliance with the Lipinski's rule of five (*Manto et al., 2018*), resulting in about 1.5 million compounds. Then, a virtual screening process with three steps of different speed

and precision including high throughput virtual screening (HTVS), standard precision (SP) and extra precision (XP) were respectively applied to the 1.5 million compounds. By setting different cutoff values for the HTVS of –5, SP of –7 and XP of –5, about 2900 compounds were obtained. The protein-ligand interaction of the crystal structures showed that Asp27 within the binding pocket is one of the most critical for forming hydrogen bond with known inhibitors. A total of 491 hits having contacts with Asp27 were filtered. After visual selection, 307 out of 491 compounds were submitted for the molecular dynamics evaluation using the previously selected DHFR crystal structure PDB 1*R*X3 as a target. Using the protocol described above, we predicted the binding affinities of all 307 compounds. With a cutoff value of less than –20kcal/mol, 40 compounds were chosen for further analysis. For those 40 compounds with 32 from the ChemDiv database and eight from the Life Chemicals database, we calculated their binding affinity for the WT and three single, three double and one triple mutants based on P21L, A26T, and L28R. Calculations predicted significant binding affinity for all compounds not only to the wild type DHFR, but also to the DHFR mutants.

Based on the principle that compounds with similar properties tend to have similar activity (*Kumar, 2011*), and to ensure that the selected compounds with chemical novelty, three types of compound similarity based on Tanimoto coefficient (*Zhang et al., 2013*) were compared between the selected hits with that of the known 183 DHFR inhibitors. The Tanimoto coefficient uses the ratio of the intersecting set to the union set as the measure of similarity when each attribute is expressed in binary. Represented as a mathematical equation:

$$T_{AB} = \frac{c}{a+b-c}$$

In this equation, individual fingerprint bits set in molecules A and B are represented by a and b, respectively; and c is the intersection set. $T_{AB}$ value ranges from 0 to 1, where 0 represents that no same bits are detected; however, 1 does not mean that the two molecules are totally identical. Two-dimensional (2D) physicochemical properties including molecular weight, AlogP, polar surface area, number of rotatable bonds, rings, aromatic rings, hydrogen acceptors and hydrogen donors were compared from the two-dimensional property's aspect. Interaction similarity based on protein-ligand interaction (*Huovinen et al., 1995*) were calculated to ensure that the selected compounds form similar interactions with the critical binding site pocket residues which can further guarantee their biological activity. ECFP4 (*Zhang et al., 2013*), an extended connectivity fingerprints which can represent the chemical diversity of compounds, was used to ensure the wide chemical space and novelty in their structures. As shown in *Figure 1—figure supplement 8*, the 2D physicochemical properties similarity were concentrated in the range of 0.6–0.8 and the protein-ligand interaction fingerprint (PLIF) similarities were mainly distributed in the range of 0.7–0.9 on the one hand, and the ECFP4 chemical similarities were only just between 0.2–0.3 on the other hand. This indicated that the selected hits were of similar 2D physicochemical property and protein-ligand binding interaction but also possessed chemical diversity compared with the known DHFR inhibitors. For detailed analysis of the 2D properties for the selected hits, the predicted values above mentioned properties including molecular weight, AlogP, polar surface area, number of rotatable bonds, rings, aromatic rings, hydrogen acceptors, and hydrogen donors were compared with that of the known DHFR inhibitors. It can be clearly seen from *Figure 1—figure supplement 9* that those 2D properties showed similar distributions with the positive controls, further proving the effectiveness of the selected hits. Finally, selected 40 hit compounds were submitted for purchase and biological activity evaluation.

## Enzymatic activity assay

All reagents and chemicals were of high quality and were procured from Sigma-Aldrich Co., USA, Amresco, or Fisher Scientific. According to our former study, wild-type (WT) *E. coli*, *C. muridarum* and *L. grayi* DHFR as well as mutant *E. coli* DHFR proteins including P21L, A26T, and L28R point mutants were expressed and purified in the same way as previously described (*Rodrigues et al., 2016*). Human DHFR was purchased from Sigma Aldrich. In an initial screen, the compounds were assayed for inhibition of DHFR catalytic activity at a single fixed concentration of 200 µM. The reaction mixtures (100 µL final volume) contained 100 µM NADPH, 30 µM dihydrofolate and 100 mM HEPES, pH 7.3. The DHFR enzyme was added to initiate the reaction (16.7 nM final concentration) and the oxidation of NADPH was monitored at 25°C by the decrease in absorbance at 340 nm. The initial velocities with product

formation less than 5% were measured and the nonenzymatic hydrolysis of NADPH was subtracted. The concentration-dependent inhibition curves were determined at variable concentration of each inhibitor. To minimize the interference at high compound concentrations, the rate of NADPH oxidation catalysed by DHFR was instead monitored by following the decrease in fluorescence at 450 nm (excitation 340 nm). Inhibition constants were determined from kinetic competition experiments performed based on varying inhibitor concentrations (*Srinivasan and Skolnick, 2015*). The curves were fit to the equation $y = \frac{100\%}{(1+I/IC_{50})}$, where I is concentration of the inhibitor and y is the percentage of catalytic activity of DHFR. The values of inhibition constant $K_i$ were obtained by fitting a competitive inhibition equation (*Krohn and Link, 2003*) from plots of activity vs. inhibitor concentration using the equation $K_i = \frac{IC_{50}}{1+\frac{[S]}{K_M}}$, where the $K_M$ values of DHFR enzyme for DHF (at saturating NADPH concentration) as measured before (*Rodrigues et al., 2016*) and [S] is the substrate concentration of DHF. All the measurements were conducted in triplicate, and the error values are indicated by standard errors. All the data were fit using the nonlinear curve-fitting subroutines OriginPro (*Seifert, 2014*).

## Bacterial growth measurements and determination of IC$_{50}$ values

Cultures in M9 minimal medium was grown overnight at 37°C and were then normalized to an OD of 0.1 using fresh medium. A new normalization to an OD = 0.1 was conducted after additional growth during 5–6 hr. Then, the M9 medium and six different concentrations of the positive control TMP and all hit compounds in the 96-well plates (1/5 dilution) were incubated. The incubation of the plates was performed at 37°C and the orbital shacking and absorbance measurements at 600 nm were taken every 30 min during 15 hr. By integration of the area under the growth curve (OD vs. time), the growth was quantified between 0 and 15 hr, as described elsewhere (*Palmer et al., 2015*). For the WT DHFR or a given mutant, the growth integrals normalized by corresponding values of growth for that same strain without the hit compounds. By fitting a logistic equation to plots of growth vs. compound concentrations, the IC$_{50}$ values were determined. The reported IC$_{50}$ values are an average based on at least three replicates and standard errors are indicated.

## Whole genome sequencing

Using single isolated colonies, whole genome sequencing for the evolved variants was performed resorting to Illumina MiSeq 2 × 150 bp paired-end configuration (Novogene). We used breqseq pipeline (*Deatherage and Barrick, 2014*), on default settings, using the *E. coli* K-12 substr. BW25113 reference genome (GenBank accession no. CP009273.1).

## Evolution experiments

Evolution of *E. coli* strains in the presence of prospective compound CD15-3 or TMP was carried out by serial passaging using an automated liquid handling system (Tecan Freedom Evo 150), a procedure similar to the one described previously (*Rodrigues and Shakhnovich, 2019*). In this setup, cultures are grown in wells of a 96-well microplate, and the optical density (600 nm) is measured periodically using a plate reader (Tecan Infinite M200 Pro). In each serial passage, the cultures are diluted with fresh growth media into the wells of the adjacent column. The growth rate measured for each culture at a given passage is compared with the growth rate determined in the absence of antibiotic and the concentration of antibiotic is increased by a factor of 2 if it exceeds (75%), otherwise maintained. To avoid excessive antibiotic inhibition, the concentration was increased only once in every two consecutive passages. This procedure forces cells to grow under sustained selective pressure at growth rates close to 50% of that of non-inhibited cells.

For plate assays, WT cells were grown for around 8 hr without any antibiotic selection. Cells were grown in M9 media supplemented with 0.8 g L$^{-1}$ glucose. The cells were thereafter plated on M9 agar plates with similar glucose and M9 salt concentraions as was in the M9 liquid media. Plating was done after 1000 fold dilution of the growing culture. The plates were having either TMP or CD15-3 as the selection inducer and their respective concentrations were 3 X of IC$_{50}$ values as is known for WT/naive cells (viz. 1.2 µg/µl for TMP and 210 µM for CD15-3). It is to be noted that six well plates were used for this study and the plated were incubated at 37°C for colonies to appear.

### Differential interference contrast (DIC)

WT cells were grown in M9 media supplemented with 0.8 gL$^{-1}$ glucose and casamino acids (mixtures of all amino acids except tryptophan) in absence and presence of CD15-3 at 42°C for incubation and 300 rpm constant shaking. Drop in DHFR activity has been associated with cellular filamentation and similar phenotype is observed under TMP treatment (*Bhattacharyya et al., 2019*). Since CD15-3 targets intracellular DHFR and soluble fraction of cellular DHFR is lower at 42°C, we chose this temperature for our imaging studies (*Bershtein et al., 2012*).

Aliquots were taken from the growing culture and they were drop casted on agar bed/blocks. These blocks were taken further processed for differential inference contrast (DIC) imaging using Zeis Discovery imaging workstation. Multiple fields were observed and scanned for a single condition type and a minimum of three replicates were used for imaging studies. Similar methods for imaging were used for evolved cell types under conditions of absence and presence of CD15-3 compound. Intellesis Module was used for analyzing DIC images. On average, around 500 cells were analyzed for computing cell length. *E. coli* cell lengths in our imaging studies were not normally distributed. Nonparametric Mann-Whitney test was therefore used to determine if the cell length distributions were significantly different upon CD15-3 treatment.

## Acknowledgements

This work was supported by NIH RO1 068670 and by the program of China Scholarship Council (CSC). We are grateful to Nicolas Chéron and Jiho Park for help with the computational setup.

## Additional information

### Funding

| Funder | Grant reference number | Author |
| --- | --- | --- |
| National Institute of General Medical Sciences | NIH RO1 068670 | Sourav Chowdhury João V Rodrigues Eugene I Shakhnovich |

The funders had no role in study design, data collection and interpretation, or the decision to submit the work for publication.

### Author contributions

Yanmin Zhang, Sourav Chowdhury, Formal analysis, Investigation, Methodology, Writing - original draft; João V Rodrigues, Conceptualization, Formal analysis, Investigation, Methodology; Eugene Shakhnovich, Conceptualization, Funding acquisition, Supervision, Writing – review and editing

### Author ORCIDs

Yanmin Zhang http://orcid.org/0000-0002-4075-7556
Sourav Chowdhury http://orcid.org/0000-0002-1148-2995
João V Rodrigues http://orcid.org/0000-0002-5605-656X
Eugene Shakhnovich http://orcid.org/0000-0002-4769-2265

### Decision letter and Author response

Decision letter https://doi.org/10.7554/eLife.64518.sa1
Author response https://doi.org/10.7554/eLife.64518.sa2

## Additional files

### Supplementary files
- Transparent reporting form
- Supplementary file 1. Compound Information.
- Supplementary file 2. Duplicated Genome Stretch.

## Data availability

All the data is made available in the paper.

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
