## [Decision Letter]

**Acceptance summary:**

Antibiotic resistance evolves rapidly in response to treatment. Resistance sometimes evolves predictably through the accumulation of specific mutations. One way to limit evolution is to develop other molecular compounds that target the WT protein and that remain efficient against these resistance mutations while exploiting the deep knowledge of the traditionally targeted pathways. Here, the authors use this strategy to identify potential compounds that help delay the evolution of antibiotic resistance in bacteria using a combination of computational, biochemical and experimental evolution approaches.

**Decision letter after peer review:**

Thank you for submitting your article "Development of evolution drugs -antibacterial compounds that block pathways to resistance" for consideration by *eLife*. Your article has been reviewed by 3 peer reviewers, one of whom is a member of our Board of Reviewing Editors, and the evaluation has been overseen by George Perry as the Senior Editor. The reviewers have opted to remain anonymous.

The reviewers have discussed the reviews with one another and the Reviewing Editor has drafted this decision to help you prepare a revised submission.

Summary:

Antibiotic resistance can evolve rapidly and thus reduces the usefulness of many drugs that are otherwise effective. The development of new drugs that could slow down adaptation to antibiotics would significantly extend the life expectancy of current treatment strategies. Here, the authors identify, through computational analyses and experimental work, new chemical compounds that target the dihydrofolate reductase of *E. coli*, the known target of the common antibiotic trimethoprim. Contrary to trimethoprim, a new lead appears to lead very slowly to resistance. The authors present their strategy and findings as a way to develop 'evolution drugs'.

Essential revisions:

The reviewers raised many points that would need to be addressed for the manuscript to be considered further. I outline here the critical elements that are essential to address.

1) The new compounds are presented as new antibiotics in the context of medical application but they are active against the human DHFR and thus lack selectivity, which is a major issue in antibiotic development. The reviewers agree that because of this, this work could not be published under the theme of inhibitor development. It could potentially be rewritten as a model system for the examination strategies to delay the onset of resistance.

2) The number of mutants sequenced is not large enough to rule out the possibility that resistance can emerge through mutations in folA. In addition, some of the mutants come from the same adapted populations, which means that they are not an independent sampling of resistance mutants. The claim that resistance mechanisms are different from that of Trimethoprim and do not involve folA itself is therefore not supported.

3) Some of the information that would have been useful for the interpretation and evaluation of the current manuscript is to be published in an upcoming publication. It would be important for the reviewers to have access to this information. This would be possible for instance by providing a link to a preprint with this other paper.

4) The concept of evolution drug needs to be better defined to make sure the reader understands its scope and what it is bringing to the field.

We also transmit the detailed comments below.

*Reviewer #1:*

I cannot comment on the sections of the manuscript related to compound search and identification.

My general comment is about what makes an evolution drug exactly. From the definition given, it is a drug that works on mutants that are resistant to other drugs. Most antibiotics that interact slightly differently with the same target, or that have other modes of action altogether would therefore be evolution drugs? Since this is the aspect of the paper that makes it very original, I would need to better understand what this concept is, what it is not, and what novelty it brings.

My second critical comment regards the evolution experiment. I could not find exactly how many independent lines were evolved to resist the novel compound and how many were sequenced. The methods mention evolution in 96 well plates but from the results, it seems that only two mutants were sequenced. I believe that this is not enough to claim that this drug will not select for mutations in fola. To be able to show this, enough independent clones have to be sampled and sequenced to show that the space of resistance mutations has been completely sampled.

My third critical comment is that it seems that CD15-3 selects for mechanisms that lead to resistance to TMP as well, which is not something that you want for an evolution drug. This means that resistance to CD15-3 brings resistance to TMP as well if I understand well. How does this fit within the concept of evolution drugs? This would seem to make it a rather inefficient one.

Detailed comments: Panels of Figure 7 would need to be better labelled.

Many barplots are used to show estimates, for instance in figure 4. This is a representation that poorly represents the data. (https://journals.plos.org/plosbiology/article?id=10.1371/journal.pbio.1002128). I recommend alternatives are used and all data points are shown in addition to error bars (which are not defined here).

*Reviewer #2:*

The authors present a computational workflow that convincingly demonstrates the design and optimization of inhibitors to a key antibiotic target, such that the development of inhibitor resistance is clearly delayed. The experimental determination of inhibition in vitro, in cells and during the course of one month under evolutionary pressure with the new compounds was convincing. Unfortunately, the new inhibitors inhibit human DHFR more strongly. They would be toxic/lethal.

The authors could examine the inclusion of negative design (against inhibition of the human enzyme) in their workflow.

Since trimethoprim resistance is at the center of this research, and that the authors aim to develop an "evolution drug", TMP-specific resistance should be described in greater depth. The specific context and reference given in the 2nd/3rd paragraphs of the introduction are: "However, due to rapid emergence of resistant mutations in DHFR, the development of drug resistance to antifolate antibiotics belonging to any of the above-mentioned classes presents a significant challenge (Huovinen et al., 1995). Both clinical and in vitro studies have shown that accumulation of point mutations in critical amino acids residues of the binding cavity represent an important mode of trimethoprim resistance. Mutations conferring resistance in bacteria to anti-DHFR compounds are primarily located in the folA locus that encodes DHFR in *E. coli* (Oz et al., 2014; Tamer et al., 2019; Toprak et al., 2012) making DHFR an appealing target to develop evolution antibiotic drugs. » This view is now known to be incomplete. Crucially, the folA-related DfrA enzyme family should be introduced (as comprehensively described in Sánchez-Osuna et al., Microbial Genomics 2020;6) because clinical TMP resistance is more complex than point mutations to FolA as described by Huovinen. The intrinsically TMP-resistant DfrB family of DHFRs should also be mentioned as being part of the problem, although this work would not directly apply to it.

As a result of the new finding that TMP resistance is very diverse and does not result only from the recent evolution of point mutations as was previously thought, the approach presented here should be qualified. In the introduction (3rd paragraph): "In this work, we developed an integrative computational modeling and biological evaluation workflow to discover novel DHFR inhibitors that are active against WT and resistant variants. » should qualify the resistant variants as being specifically directed to the point-mutated variants only, not all resistant variants that are genetically diverse.

It thus follows that the premise, as currently stated, is somewhat misleading. The introduction discusses clinical observation of antibiotic evolution. However, the point mutations under study are not particularly relevant to the natural evolution of TMP resistance (prior to introduction of TMP), or even its clinical evolution. This is clearly shown by the modest TMP resistance offered (Figure 2C). As described under Methods, section "Construction of binding affinity prediction model": "An evolutionary study (Oz et al., 2014; Toprak et al., 2012) of TMP resistance found that three key resistance mutations P21L, A26T, L28R, and their combinations constitute a set that recurrently occurred in two out of five independent evolution experiments, and their order of fixation in both cases was similar." This set of mutations constitutes a simple model system of in vitro evolution. This does not remove from the success of the inhibitor discovery and delay of onset of resistance, but it must be made clear that the premise is limited to in vitro evolution as a model of natural evolution, from the Abstract and throughout the body of the work.

Following Figure 1: "…*Listeria* grayi (L. grayi) and Chlamydia muridarum (C. muridarum) again showing highly significant correlation between predicted and experimental values (Figure S7), demonstrating broad predictive power of the method". To highlight the relevance of the study, the choice of *Listeria* grayi and Chlamydia muridarum should be rationalized. The native resistance of L. grayi should be expressed; the choice of C. muridarum is not so clear. Importantly, what is their evolutionary distance with *E. coli* FolA? Therefore, do they demonstrate broad predictive power, as stated, and does it suggest that they are broadly efficient potential antibacterial leads (above Figure 3, regarding inhibition by CD15 and CD17)?

The authors chose to characterize the in vivo IC50 of the compounds. Since the role of compounds is to inhibit bacterial growth completely, minimal inhibitory concentration (MIC) and minimal bactericidal concentration (MBC) might be more relevant in this context and could be determined for the most interesting compounds, and different targets.

Suggestions and typographical:

– A recent review on 'classical' searches for inhibitors could be cited in the 2nd paragraph of the introduction: Wróbel, Arciszewska, Maliszewski and Drozdowska The Journal of Antibiotics volume 73, pages5-27(2020)

– Confusion in describing the activity test in vitro: in the body the authors describe fluorescence at 340 nm (Figure 2) whereas Methods describe the decrease in absorbance at 340 nm. Figure 2A: Which is the curve of TMP?

– Table 1: significant digits on values are not uniform – should be two.

– *E. coli* and in vitro are not routinely italicized.

– CD15-3 or CD15.3.

– Table 1: Ki of TMP should be included for comparative purposes.

– Table 2: IC50 values should have no more than 2 significant digits. TMP should have STD.

– Figure 7C: Concentration in µM, not µm.

– Figure 9. The order of the panels should be the same as the order of the *E. coli* variants presented in panel E. In each DIC image, there should be a size bar (ex. 5 µm).

*Reviewer #3:*

In this manuscript, the authors present a strategy to identify new antibiotic compounds that block DHFR -- the target of trimethoprim -- in a way that should still work in DHFR variants carrying common trimethoprim resistance mutations. Their strategy consists of a computational approach to select promising candidates from known databases, followed by an experimental optimization and validation of the identified compounds. Interestingly, the first computational steps already narrowed down large chemical databases into a few relevant compounds, which are then shown to be efficient DHFR inhibitors in the experimental tests. Moreover, evolution of resistance to the resulting compound is shown to be much slower than resistance to trimethoprim.

The authors should address the following points (roughly in order of importance):

– An upcoming publication, in which a second molecular target of compound CD15-3 is revealed, is mentioned several times. It is slightly frustrating that these results, which may be key to understanding some of the phenomena reported in the present work (see below), are not included in this work. Why can this information not be included in the current article?

– Apart from non-synonymous mutations in folA, a common alternative way to evolve trimethoprim resistance is to overexpress DHFR. This happens in typical evolution experiments and in the clinic, for example through mutations that affect the folA promoter (see e.g. Toprak et al., Nature Genetics, 2012); this could also happen via duplication (or further amplification) of the folA gene. Figure 6 shows that, indeed, for the CD15-3 compound, overexpressing DHFR would actually allow the cell to recover from the effect of the compound. It is thus quite striking that no overexpression of DHFR seems to occur in the evolution experiments shown in figure 7A, where no mutation in folA fixes. The authors should comment on this in more detail. Could the other (mysterious) target of the drug explain this?

– How does the absolute value of the IC50 of the new compound compare with that of trimethoprim?

– The manuscript has a considerable amount of technical details in the main text, which makes it unnecessarily long and sometimes difficult to follow the main ideas. A shorter main text with fewer figures would considerably increase the clarity and accessibility of this work for non-experts (the details could be shifted to a supplement or appendix).

– In figure 3, a more explicative caption title would help the reader to quickly grasp the main conclusion from this figure.

– It is not completely clear in how far the approach presented in this work can be extended to other antibiotics and their resistance targets. It would be interesting to discuss this in more detail in the manuscript.

[Editors' note: further revisions were suggested prior to acceptance, as described below.]

Thank you for resubmitting your work entitled "Development of antibacterial compounds that block evolutionary pathways to resistance" for further consideration by *eLife*. Your revised article has been evaluated by George Perry (Senior Editor) and a Reviewing Editor.

The manuscript has been improved but there are some remaining issues that need to be addressed, as outlined below:

The reviewers and I appreciate the numerous modifications that were made to the manuscript. You will find below some comments from the reviewers. I will not reproduce them here because I believe they are all important and would need to be addressed. Also, I appreciated the additional experiment to isolate resistant mutants (Line 464). However, all mutants isolated come from the same culture, so it is not possible to know whether they are independent mutants. This experiment suffers from the same limitation as the previous experiment. Mutants from independent cultures would need to be isolated for them to be truly independent. The claim that resistance to CD15-3 does not evolve through mutations in folA would need to be expressed as < resistance to CD15-3 is less likely to evolve through mutations in folA> rather than <does not evolve through mutations in folA>. Saying that it does not happen would require a comprehensive analysis of mutants. The first comment of reviewer #3 also relates to this point.

*Reviewer #2:*

The authors made important improvements to the manuscript, addressing many key issues. In particular, demonstration of lack of inhibition of the human DHFR by CD15-3 is encouraging. This point should be highlighted by including the maximal concentration tested in footnote (b) of Table 1. The overall strategy is now better contextualized and has been strengthened by the additional data included in this version of the manuscript.

It is indeed interesting that (lines 712…) "the CD15-3 inhibitory activity extends beyond the DHFR variants that were initially selected as targets for structure-based design and that "unplanned" D27E and W30 mutants get inhibited with WT like efficacy." The design strategy included known single/double mutations at positions 21, 26 and 28, indicated in Table 1; in the case of CD15-3 only one point mutant Ki was assessed. The Ki of CD15-3 for the other mutants used in the design strategy (Table 1) must be included, otherwise, there is no direct validation of the strategy. This is made obvious by the unclear conclusions of the cellular inhibition results, where the authors go to great lengths to attempt to isolate the effect of the inhibitors on DHFR in the cells (Section 'Target validation in vivo'). Therefore, validation of the strategy on the direct targets used in the design, the mutant DHFRs, should be done.

*Reviewer #3:*

The authors made a substantial effort and addressed most critical points in the revision. It is also very helpful that they made the preprint of the related article available online. However, there are a few remaining points that still need to be addressed:

– In their response, the authors speculate that, because of the second target of the compound, overexpression of folA would be a less optimal evolutionary solution. This is vague and still not clear because the data show that overexpression of folA is an easy way to increase resistance, irrespective of any other drug target. Moreover, in the second article they show that overexpressing the other target (folK) leads to even higher resistance to the drug. Thus, there is a similar issue: Why do solutions that overexpress this target (which should be easily accessible by promoter mutations or gene amplification) do not appear in the evolution experiments? I understand that it may not be possible to elucidate this in a reasonable time frame, but it should be clearly stated and discussed in detail in the manuscript (not just in the response to the reviewer comments). This point is important because it changes the main take-home message from "It is not possible to evolve resistance to the new compound" to "There are the same straightforward ways to evolve resistance to the new compound as for trimethoprim but (for unknown reasons) they are not followed in evolution experiments."

– Another point that is addressed in the response to reviewers but still needs to be transparently stated and discussed in the revised manuscript is how the IC50 of the new compound compares to that of trimethoprim. The most helpful would be to show the dose-response curves for the new compound and for trimethoprim in the same plot with absolute concentrations on the x-axis. A similar plot that directly compares the effect of overexpressing folA on both dose-response curves would also help.

---

## [Author Response]

Essential revisions:The reviewers raised many points that would need to be addressed for the manuscript to be considered further. I outline here the critical elements that are essential to address.1) The new compounds are presented as new antibiotics in the context of medical application but they are active against the human DHFR and thus lack selectivity, which is a major issue in antibiotic development. The reviewers agree that because of this, this work could not be published under the theme of inhibitor development. It could potentially be rewritten as a model system for the examination strategies to delay the onset of resistance.

We thank the reviewers for pointing out this issue. Indeed any potential antibiotic lead has to exhibit selective activity against the bacterial protein and be inactive against its human orthologue. However the activity data we had earlier presented in our manuscript was of CD15, the parent compound for our lead CD15-3. The work essentially focuses on the potential of CD15-3 as an antibiotic lead. We assessed the CD15-3 activity against human DHFR and found that, unlike parent compound CD15, it does not inhibit human DHFR (Figure 4—figure supplement 1 in the manuscript) and hence shows selectivity against only bacterial DHFR.

CD15-3 structurally differs from CD15 with a naphthalene ring instead of a dimethoxy substituted benzene ring which can potentially explain loss of its with human DHFR. By comparison of the crystal structures of *E. coli* DHFR (PDB 1RX3) and human DHFR (PDB 1KMS), we found there are significant sequence differences between them in the ligand-binding domain. Different pairs of amino acids between *E. coli* DHFR and human DHFR include Met20: Leu22, Asp27: Glu30, Leu28: Phe31, Trp30: Tyr33, Ile94: Val115 (Figure 4—figure supplement 2 in the manuscript). From the binding of CD15 and CD15-3 against the two proteins, the loss of inhibition on human DHFR for CD15-3 is probably due to the loss of the pi-pi interaction with Phe31. As for the *E. coli* DHFR, the corresponding amino acid is Leu28, and it does not affect binding CD15-3.

2) The number of mutants sequenced is not large enough to rule out the possibility that resistance can emerge through mutations in folA. In addition, some of the mutants come from the same adapted populations, which means that they are not an independent sampling of resistance mutants. The claim that resistance mechanisms are different from that of Trimethoprim and do not involve folA itself is therefore not supported.

We thank the reviewers for this critical comment. Indeed, in our last submission we presented limited evolution trajectories. In our current submission we have introduced two additional evolution trajectories. We carried out the same evolution experiments following the protocols earlier mentioned and allowed bacterial cells to evolve in presence of CD15-3 for over a month. Upon sequencing the “CD15-3-evolved” batch we did not observe any mutation in the folA locus. Further these evolved batches again showed a little more than two fold increases in IC_50_ compared to the naïve cells, much lesser than the enhancement observed in the “TMP-evolved” batch (8 to 200 fold compared to the WT/naive).

In addition, we carried out an alternative strategy of “one-shot” selection of colonies that are capable of growth in presence of TMP or CD15-3. Again, colonies grown in presence of CD15-3 do not show variation in the folA locus (see new Table 3 and accompanying discussion) and show a lower increase in resistance when compared to TMP.

3) Some of the information that would have been useful for the interpretation and evaluation of the current manuscript is to be published in an upcoming publication. It would be important for the reviewers to have access to this information. This would be possible for instance by providing a link to a preprint with this other paper.

We thank the reviewers for this point. We provided a link to the recently submitted Biorxiv pre-print of the paper discussing CD15-3 intra-cellular mode of action in the relevant parts of the manuscript.

4) The concept of evolution drug needs to be better defined to make sure the reader understands its scope and what it is bringing to the field.

We appreciate this suggestion. Earlier we had provided a brief overview behind the concept of “evolution-drug” in the introduction section. We had re-formatted that section and introduced some additional information also in the discussion to provide a better conceptual background. Corresponding changes in the Introduction and Discussion are marked in blue.

We also transmit the detailed comments below.Reviewer #1:I cannot comment on the sections of the manuscript related to compound search and identification.My general comment is about what makes an evolution drug exactly. From the definition given, it is a drug that works on mutants that are resistant to other drugs. Most antibiotics that interact slightly differently with the same target, or that have other modes of action altogether would therefore be evolution drugs? Since this is the aspect of the paper that makes it very original, I would need to better understand what this concept is, what it is not, and what novelty it brings.

We thank the reviewer for this point. In our earlier submission we had briefly mentioned what “evolution-drugs” mean and we have re-formatted that section. Also we have clarified the concept in the direction which reviewer had mentioned.

Conceptually “evolution drug” is a compound which stresses pathogens or cancer cells and suppresses their evolutionary pathways to escape. Antibiotic resistance or drug resistance in general is an evolutionary selection of phenotype which evades the drug action by deploying multiple molecular level escape strategies. These strategies could be as general as efficient efflux of the drug or specific like mutation of the target locus resulting in the target protein modification which would eliminate binding of the antibiotic molecule. In our study we designed a strategy to have antibiotic leads which would not only interact with the WT protein but would interact with a subset of target protein escape variants. Since compounds designed like this can potentially block the evolutionary escape routes of bacterial cells growing under antibiotic selection, the strategy could be potentially promising to block the emergence of “antibiotic-resistant” bacterial phenotypes.

My second critical comment regards the evolution experiment. I could not find exactly how many independent lines were evolved to resist the novel compound and how many were sequenced. The methods mention evolution in 96 well plates but from the results, it seems that only two mutants were sequenced. I believe that this is not enough to claim that this drug will not select for mutations in fola. To be able to show this, enough independent clones have to be sampled and sequenced to show that the space of resistance mutations has been completely sampled.

We thank the reviewer for pointing out this issue. In our previous submission we had presented one evolution trajectory/line. Two colonies obtained after plating the evolved batch were selected and sequenced. In our current submission we have added two more evolution trajectories and we did not observe any mutation in the folA locus. Also we have carried out a parallel plate based study (new Table 3 and accompanying discussion) and found the colonies formed on CD15-3 plate do not have mutation in the folA locus compared to the ones on TMP plate which had mutations in the folA locus.

My third critical comment is that it seems that CD15-3 selects for mechanisms that lead to resistance to TMP as well, which is not something that you want for an evolution drug. This means that resistance to CD15-3 brings resistance to TMP as well if I understand well. How does this fit within the concept of evolution drugs? This would seem to make it a rather inefficient one.

We thank the reviewer for this point and appreciate the observation. It is true that CD15-3 escape/evolved variant offered moderate resistance to TMP as was evident from 3 fold enhanced IC_50_ compared to that of the naïve cells. But this resistance could be attributed to the transporters and pumps which constituted around 38% of the genes in the duplicated segment. This is a very generic mechanism by which evolving bacterial cells neutralize the toxicity of the drugs. We showed that a 30+ days of evolution under CD15-3 leads to the modest increase in IC_50_ but it does not involve any mutation in the folA locus. It is interesting to note that CD15-3 unlike TMP was not only designed to inhibit WT DHFR but also some prominent mutants of DHFR which emerge under TMP selection. Since CD15-3 significantly constraints the escape routes associated with folA locus mutation, it is likely that the resistance would emerge by generic mechanisms as mediated by efflux in the current variants.

Further CD15-3 significantly delayed the onset of resistance as the IC_50_ of the evolved variants were just 2 times compared to the naïve cells. During the same timescale of evolution, the IC_50_ of the TMP escape variants went to 8 to 200 times higher compared to that of the naïve cells.

Detailed comments: Panels of Figure 7 would need to be better labelled.Many barplots are used to show estimates, for instance in figure 4. This is a representation that poorly represents the data. (https://journals.plos.org/plosbiology/article?id=10.1371/journal.pbio.1002128). I recommend alternatives are used and all data points are shown in addition to error bars (which are not defined here).

We thank the reviewer for pointing out these two points. We changed the Figure 7 in the current submission.

The reviewer suggested us to change our bar graph in Figure 4 to an alternative one. However, when we generated the bar graph in Figure 4, we considered three replicate tests for the inhibition ratios and then took an average of the ratios and converted them into average ki value and its standard error.

Reviewer #2:The authors present a computational workflow that convincingly demonstrates the design and optimization of inhibitors to a key antibiotic target, such that the development of inhibitor resistance is clearly delayed. The experimental determination of inhibition in vitro, in cells and during the course of one month under evolutionary pressure with the new compounds was convincing. Unfortunately, the new inhibitors inhibit human DHFR more strongly. They would be toxic/lethal.The authors could examine the inclusion of negative design (against inhibition of the human enzyme) in their workflow.

We thank the reviewer for pointing out this concern. Table 1 in our earlier submission primarily documented the CD15 activity profile. It is to be noted that CD15-3 which we present as the potential antibiotic lead is a derivative of CD15. We carefully measured the activity of CD15-3 against human DHFR and found that CD15-3 does not show any inhibitory activity (see Figure 4—figure supplement 1 in the manuscript). Also in the current submission we have incorporated a table showing the in vitro activity profile of CD15-3.

Indeed in our earlier submission when we tried to find novel *E. coli* DHFR inhibitors, their activity against human DHFR were not considered in the computational search. We only used the *E. coli* DHFR crystal structure for the virtual screening. Later we tested CD15-3 against human DHFR and found that CD15-3 unlike CD15 did not inhibit human DHFR activity.

We thank the reviewer for the valuable suggestion on negative design. We would include a negative design in later steps of the screening workflow in our future efforts and calculate the binding activity of the hit compounds against human DHFR and remove those compounds with high affinity against human DHFR.

But in this work, since the hits were already screened, so we would like to try to use this method to further optimize the hits to improve their activity. Nevertheless, we calculated the binding free energy of TMP against *E. coli* WT and L28R DHFR and human DHFR by using molecular dynamics and MM/PBSA. We found that the binding free energy for TMP is lower on the human DHFR than on *E. coli* WT and L28R DHFR. This can partly explain why TMP is highly potent against microbial DHFRs than the human DHFR. As can be seen from Author response table 1, CD15-3 is much less active against human DHFR than that of *E. coli* WT or L28R DHFR.

**Author response table 1. sa2table1:** Calculated binding free energy of TMP, CD15 and CD15-3 against *E. coli* WT and L28R DHFR and human DHFR, respectively.

Molecular Dynamics ^a^	MM/PBSA (kcal/mol)		
	1st	2nd	Average
EcDHFR^b^ WT_TMP	-25.24	-26.23	-25.74
EcDHFR L28R_TMP	-24.44	-21.57	-23.01
hDHFR^c^_TMP	-20.49	-24.04	-22.27
EcDHFR WT_CD15	-33.22	-34.30	-33.76
EcDHFR L28R_CD15	-27.68	-28.61	-28.15
hDHFR_CD15	-31.75	-32.16	-31.96
EcDHFR WT_CD15-3	-32.38	-31.80	-32.09
EcDHFR L28R_CD15-3	-31.60	-34.29	-32.95
hDHFR_CD15-3	-20.02	-11.72	-15.87

^a^ Two replicates of 10ns molecular dynamics simulation were conducted for each system, and the last 2ns trajectory were used to calculate the binding free energy of each simulation and obtain the average. ^b^EcDHFR represents *E. coli* DHFR. ^c^hDHFR represents human DHFR.

Since trimethoprim resistance is at the center of this research, and that the authors aim to develop an "evolution drug", TMP-specific resistance should be described in greater depth. The specific context and reference given in the 2nd/3rd paragraphs of the introduction are: "However, due to rapid emergence of resistant mutations in DHFR, the development of drug resistance to antifolate antibiotics belonging to any of the above-mentioned classes presents a significant challenge (Huovinen et al., 1995). Both clinical and in vitro studies have shown that accumulation of point mutations in critical amino acids residues of the binding cavity represent an important mode of trimethoprim resistance. Mutations conferring resistance in bacteria to anti-DHFR compounds are primarily located in the folA locus that encodes DHFR in *E. coli* (Oz et al., 2014; Tamer et al., 2019; Toprak et al., 2012) making DHFR an appealing target to develop evolution antibiotic drugs. » This view is now known to be incomplete. Crucially, the folA-related DfrA enzyme family should be introduced (as comprehensively described in Sánchez-Osuna et al., Microbial Genomics 2020;6) because clinical TMP resistance is more complex than point mutations to FolA as described by Huovinen. The intrinsically TMP-resistant DfrB family of DHFRs should also be mentioned as being part of the problem, although this work would not directly apply to it.As a result of the new finding that TMP resistance is very diverse and does not result only from the recent evolution of point mutations as was previously thought, the approach presented here should be qualified. In the introduction (3rd paragraph): "In this work, we developed an integrative computational modeling and biological evaluation workflow to discover novel DHFR inhibitors that are active against WT and resistant variants. » should qualify the resistant variants as being specifically directed to the point-mutated variants only, not all resistant variants that are genetically diverse.It thus follows that the premise, as currently stated, is somewhat misleading. The introduction discusses clinical observation of antibiotic evolution. However, the point mutations under study are not particularly relevant to the natural evolution of TMP resistance (prior to introduction of TMP), or even its clinical evolution. This is clearly shown by the modest TMP resistance offered (Figure 2C). As described under Methods, section "Construction of binding affinity prediction model": "An evolutionary study (Oz et al., 2014; Toprak et al., 2012) of TMP resistance found that three key resistance mutations P21L, A26T, L28R, and their combinations constitute a set that recurrently occurred in two out of five independent evolution experiments, and their order of fixation in both cases was similar." This set of mutations constitutes a simple model system of in vitro evolution. This does not remove from the success of the inhibitor discovery and delay of onset of resistance, but it must be made clear that the premise is limited to in vitro evolution as a model of natural evolution, from the Abstract and throughout the body of the work.

We thank the reviewer for this extensive suggestion on better putting the perspective of resistance against TMP. It is true that the TMP resistance under clinical setting is far more complex and the recent studies cited are extremely insightful in that direction. We have re-drafted the sections pointed and included the studies suggested. In particular we note in lines 94-95 “In our current study we focus on a simplified model system of in vitro evolution whereby resistant phenotypes emerge from point mutations in the folA locus.”

Following Figure 1: "…Listeria grayi (L. grayi) and Chlamydia muridarum (C. muridarum) again showing highly significant correlation between predicted and experimental values (Figure S7), demonstrating broad predictive power of the method". To highlight the relevance of the study, the choice of Listeria grayi and Chlamydia muridarum should be rationalized. The native resistance of L. grayi should be expressed; the choice of C. muridarum is not so clear. Importantly, what is their evolutionary distance with *E. coli* FolA? Therefore, do they demonstrate broad predictive power, as stated, and does it suggest that they are broadly efficient potential antibacterial leads (above Figure 3, regarding inhibition by CD15 and CD17)?

We thank the reviewer for the important point raised. The reason why we also include *Listeria* grayi (L. grayi) and Chlamydia muridarum (C. muridarum) for verifying the binding affinity prediction linear model is as follows.

In one of the former study from the group (Rodrigues et al., 2016) depicting broader view of the biophysical nature of the fitness landscape of antibiotic resistance we transformed *E. coli* cells with plasmids overexpressing DHFR from two mesophilic bacteria viz. C. muridarum and L. grayi, sharing 26% and 36% sequence identity with *E. coli* DHFR, respectively and carrying all combinations of the three key mutations in the loci corresponding to the three escape mutations in *E. coli* DHFR that we study here (see sequence alignment in Figure S3).

“Likewise in this work apart from *E. coli* DHFR we purified all variants of the mutant orthologs from *C. Muridarum* and *L. grayi,* characterized their biophysical properties, and compared them with *E. coli* DHFR mutants (Figures S2 and S4). We observed a similar trade-off between catalytic activity and TMP binding (Figure 1F, Inset). ” (from Rodrigues at al 2016).

Using DHFRs from two additional sources allowed us to significantly extend the dynamic range of biophysical parameters to provide a comprehensive biophysical mapping of the fitness landscape. Also, we obtained the inhibition data for TMP against WT and three point mutant DHFR from *E. coli*, *C. Muridarum* and *L. grayi* respectively, which enabled us to construct the binding affinity prediction linear models. And our results (Figure 1, Figure 1—figure supplement 4, 6 and 7) show that highly significant correlations between predicted and experimental values are obtained for all three species, demonstrating broad predictive power of the method.

The authors chose to characterize the in vivo IC50 of the compounds. Since the role of compounds is to inhibit bacterial growth completely, minimal inhibitory concentration (MIC) and minimal bactericidal concentration (MBC) might be more relevant in this context and could be determined for the most interesting compounds, and different targets.

For testing the antibacterial property of the compounds described we measured the effective concentrations of the compounds at which half of the growth rate compared (IC_50_) to the zero antibiotic concentration is reached. Since the compounds had limited solubility at high concentrations, going beyond a certain compound concentration to derive minimal inhibitory or batericidal concentration was not possible.

Suggestions and typographical:– A recent review on 'classical' searches for inhibitors could be cited in the 2nd paragraph of the introduction: Wróbel, Arciszewska, Maliszewski and Drozdowska The Journal of Antibiotics volume 73, pages5-27(2020)

This article has been cited in the introduction.

– Confusion in describing the activity test in vitro: in the body the authors describe fluorescence at 340 nm (Figure 2) whereas Methods describe the decrease in absorbance at 340 nm. Figure 2A: Which is the curve of TMP?

We used both absorption and fluorescence at 450 nm (excitation 340 nm) to assay the inhibition of DHFR catalytic activity. The initial screen of the 40 compounds was made by measuring the rate of NADPH oxidation monitored the decrease in absorbance at 340 nm. For the concentration dependence studies we measured NADPH oxidation by monitoring the fluorescence at 450 nm (excitation at 340nm), to avoid interferences at high compound concentrations. We clarified this aspect in the revised version. Figure 2A represents an example of in vitro kinetic assay of DHFR catalytic activity. This figure illustrates the inhibition of DHFR at by the hit compound CD15 at different concentrations. The bottom two lines are controls without CD15 but using equal amount of DMSO instead.

– Table 1: significant digits on values are not uniform – should be two.

The significant digits on values from Table 1 have been set to two.

– *E. coli* and in vitro are not routinely italicized,

*E. coli* and in vitro are all italicized in the whole manuscript.

– CD15-3 or CD15.3.

We keep CD15-3 for the compound ID. Typographic errors of CD15.3 were corrected.

– Table 1: Ki of TMP should be included for comparative purposes.

The Ki values for TMP were added to Table 1.

– Table 2: IC50 values should have no more than 2 significant digits. TMP should have STD.

This was corrected.

– Figure 7C: Concentration in µM, not µm.

This was corrected.

– Figure 9. The order of the panels should be the same as the order of the *E. coli* variants presented in panel E. In each DIC image, there should be a size bar (ex. 5 µm).

This was corrected.

Reviewer #3:In this manuscript, the authors present a strategy to identify new antibiotic compounds that block DHFR -- the target of trimethoprim -- in a way that should still work in DHFR variants carrying common trimethoprim resistance mutations. Their strategy consists of a computational approach to select promising candidates from known databases, followed by an experimental optimization and validation of the identified compounds. Interestingly, the first computational steps already narrowed down large chemical databases into a few relevant compounds, which are then shown to be efficient DHFR inhibitors in the experimental tests. Moreover, evolution of resistance to the resulting compound is shown to be much slower than resistance to trimethoprim.The authors should address the following points (roughly in order of importance):– An upcoming publication, in which a second molecular target of compound CD15-3 is revealed, is mentioned several times. It is slightly frustrating that these results, which may be key to understanding some of the phenomena reported in the present work (see below), are not included in this work. Why can this information not be included in the current article?

We thank the reviewer for the point mentioned. In our current submission we have provided the link to our pre-print “(Chowdhury et al., 2021)” at relevant places in the manuscript.

– Apart from non-synonymous mutations in folA, a common alternative way to evolve trimethoprim resistance is to overexpress DHFR. This happens in typical evolution experiments and in the clinic, for example through mutations that affect the folA promoter (see e.g. Toprak et al., Nature Genetics, 2012); this could also happen via duplication (or further amplification) of the folA gene. Figure 6 shows that, indeed, for the CD15-3 compound, overexpressing DHFR would actually allow the cell to recover from the effect of the compound. It is thus quite striking that no overexpression of DHFR seems to occur in the evolution experiments shown in figure 7A, where no mutation in folA fixes. The authors should comment on this in more detail. Could the other (mysterious) target of the drug explain this?

We thank the reviewer for this observation. It is true that a common way to evolve resistant against anti-folates like TMP is to overexpress DHFR. We did not observe any overexpression DHFR due to upstream mutation and it could be because that CD15-3 has an additional non-DHFR intra-cellular target (Chowdhury et al., 2021). This makes resistance by gene overexpression a less evolutionarily optimal solution. Resistance by overexpressing the pumps constituting the gene-duplicated segment provides a much more generic solution leading to CD15-3 resistance.

– How does the absolute value of the IC50 of the new compound compare with that of trimethoprim?

It is about 100 fold higher but the relevant comparison is with escape variants such as L28R where it is 3 fold lower than TMP.

– The manuscript has a considerable amount of technical details in the main text, which makes it unnecessarily long and sometimes difficult to follow the main ideas. A shorter main text with fewer figures would considerably increase the clarity and accessibility of this work for non-experts (the details could be shifted to a supplement or appendix).

*eLife* does not allow supplementary text per their style. Therefore, we had to incorporate many important details in main text but tried to do our best to make the paper modular so that a reader not interested in this or that detail of the multitool approach can omit them without sacrificing general understanding of the flow of logic and results.

– In figure 3, a more explicative caption title would help the reader to quickly grasp the main conclusion from this figure.

The caption title of Figure was revised to “Figure 3- The inhibition effect of compounds CD15 and CD17 against WT and mutant DHFR from different species.”

– It is not completely clear in how far the approach presented in this work can be extended to other antibiotics and their resistance targets. It would be interesting to discuss this in more detail in the manuscript.

We thank the reviewer this point. We have incorporated some of the relevant details in the current submission.

In the present work we considered *E. coli* DHFR as the drug target and used structure- and ligand-based drug design methods to screen compound databases in an attempt to find hits that can not only inhibit WT DHFR but would inhibit TMP-escape mutant forms of DHFR. In the computational workflow, a variety of computational methods were carried out for virtual screening, such as molecular docking to simulate drug target interactions and screen large compound libraries, molecular dynamics combined with free energy calculations to construct drug target affinity prediction models, We obtained potent inhibitors that were inhibitory to WT and three TMP escape mutants. We found that CD15-3 was also effective in inhibiting some other TMP escape mutants which emerged from our *in-vitro* evolution experiment.

When this method is applied to other targets with an aim to address the problem of drug resistance, we can consider building a predictive model from known inhibitors against resistant mutations, which may contribute to the drug discovery process. Some of the related references are shown as follows.

1. Perryman A L , Lin J H , Mccammon J A. HIV-1 protease molecular dynamics of a wild-type and of the V82F/I84V mutant: Possible contributions to drug resistance and a potential new target site for drugs[J]. Protein Science, 2004.

2. Zhang Y , Shen H , Zhang M , et al. Exploring the Proton Conductance and Drug Resistance of BM2 Channel through Molecular Dynamics Simulations and Free Energy Calculations at Different pH Conditions[J]. Journal of Physical Chemistry B, 2013, 117(4):982-8.

3. Hao, Zhang, Xinheng, et al. How does the novel T315L mutation of breakpoint cluster region-abelson (BCR-ABL) kinase confer resistance to ponatinib: a comparative molecular dynamics simulation study [J]. Journal of biomolecular structure and dynamics, 2019:1-12.

4. Avilaq B A , Baday S. Investigation of Drug Resistance Mechanisms for Antiandrogen Prostate Cancer Drug Enzalutamide using Molecular Dynamics Simulations[J]. Biophysical Journal, 2020, 118(3):194a.

5. Djoumbou F Y. Molecular Dynamics Based Prediction of HIV-1 Drug Resistance. 2008.

6. Jian J T , Sun T G , Wei Z C , et al. Molecular Dynamics Simulation of HIV1 gp41 and the N554D/S649A Double Mutation for Drug Resistance to Enfuvirtide. IEEE, 2009.

7. Zhou X , Du J , Zhou X , et al. Computer-aided design of PVR mutants with enhanced binding affinity to TIGIT[J]. Cell Communication and Signaling, 2021, 19(1).

8. Sun L , Zhou Y X , Wang X D , et al. Ab initio molecular dynamics and materials design for embedded phase-change memory[J]. npj Computational Materials, 2021.

Reference:

Chowdhury, S., Zielinski, D.C., Dalldorf, C., Rodrigues, J.V., Palsson, B., and Shakhnovich, E. (2021). A systems-guided approach to discover the intracellular target of a novel evolution-drug lead. bioRxiv, 2021.2005.2017.444532.

Rodrigues, J.V., Bershtein, S., Li, A., Lozovsky, E.R., Hartl, D.L., and Shakhnovich, E.I. (2016). Biophysical principles predict fitness landscapes of drug resistance. Proceedings of the National Academy of Sciences *113*, E1470-E1478.

[Editors' note: further revisions were suggested prior to acceptance, as described below.]

The manuscript has been improved but there are some remaining issues that need to be addressed, as outlined below:The reviewers and I appreciate the numerous modifications that were made to the manuscript. You will find below some comments from the reviewers. I will not reproduce them here because I believe they are all important and would need to be addressed. Also, I appreciated the additional experiment to isolate resistant mutants (Line 464). However, all mutants isolated come from the same culture, so it is not possible to know whether they are independent mutants. This experiment suffers from the same limitation as the previous experiment. Mutants from independent cultures would need to be isolated for them to be truly independent. The claim that resistance to CD15-3 does not evolve through mutations in folA would need to be expressed as < resistance to CD15-3 is less likely to evolve through mutations in folA> rather than <does not evolve through mutations in folA>. Saying that it does not happen would require a comprehensive analysis of mutants. The first comment of reviewer #3 also relates to this point.

Thank you for pointing this out. Now we included the results from two more independent starter-cultures. As before, we did not observe any mutation in the folA locus under the conditions of “one-shot” CD15-3 selection.

Further, we changed/rephrased sections expressing “does not evolve through mutations in folA” to “resistance to CD15-3 is less likely to evolve through mutations in folA” (lines 712 and 719 to 723) as correctly pointed out by the editor.

Reviewer #2:The authors made important improvements to the manuscript, addressing many key issues. In particular, demonstration of lack of inhibition of the human DHFR by CD15-3 is encouraging. This point should be highlighted by including the maximal concentration tested in footnote (b) of Table 1. The overall strategy is now better contextualized and has been strengthened by the additional data included in this version of the manuscript.It is indeed interesting that (lines 712…) "the CD15-3 inhibitory activity extends beyond the DHFR variants that were initially selected as targets for structure-based design and that "unplanned" D27E and W30 mutants get inhibited with WT like efficacy." The design strategy included known single/double mutations at positions 21, 26 and 28, indicated in Table 1; in the case of CD15-3 only one point mutant Ki was assessed. The Ki of CD15-3 for the other mutants used in the design strategy (Table 1) must be included, otherwise, there is no direct validation of the strategy. This is made obvious by the unclear conclusions of the cellular inhibition results, where the authors go to great lengths to attempt to isolate the effect of the inhibitors on DHFR in the cells (Section 'Target validation in vivo'). Therefore, validation of the strategy on the direct targets used in the design, the mutant DHFRs, should be done.

We thank the reviewer for pointing out this oversight. In our current submission we added the Ki of CD15-3 for two additional mutants’ viz. P21L and A26T, (Table 1) which were included in our original CD15-3 design scheme.

Reviewer #3:The authors made a substantial effort and addressed most critical points in the revision. It is also very helpful that they made the preprint of the related article available online. However, there are a few remaining points that still need to be addressed:– In their response, the authors speculate that, because of the second target of the compound, overexpression of folA would be a less optimal evolutionary solution. This is vague and still not clear because the data show that overexpression of folA is an easy way to increase resistance, irrespective of any other drug target. Moreover, in the second article they show that overexpressing the other target (folK) leads to even higher resistance to the drug. Thus, there is a similar issue: Why do solutions that overexpress this target (which should be easily accessible by promoter mutations or gene amplification) do not appear in the evolution experiments? I understand that it may not be possible to elucidate this in a reasonable time frame, but it should be clearly stated and discussed in detail in the manuscript (not just in the response to the reviewer comments). This point is important because it changes the main take-home message from "It is not possible to evolve resistance to the new compound" to "There are the same straightforward ways to evolve resistance to the new compound as for trimethoprim but (for unknown reasons) they are not followed in evolution experiments."

We thank the reviewer for his critical comments. We have rephrased the relevant sections in the manuscript as per the suggestion. (Lines 101-102, 712, 720-724).

– Another point that is addressed in the response to reviewers but still needs to be transparently stated and discussed in the revised manuscript is how the IC50 of the new compound compares to that of trimethoprim. The most helpful would be to show the dose-response curves for the new compound and for trimethoprim in the same plot with absolute concentrations on the x-axis. A similar plot that directly compares the effect of overexpressing folA on both dose-response curves would also help.

We thank the reviewer for this point. In figure 4E we have provided a comparative bar-plot of IC_50_ values for TMP and CD15-3. Since TMP shows efficacy at a nano-molar level compared to micro-molar levels of CD15-3, a cumulative dose response plot is otherwise hard to plot.